# Microglia are essential for tissue contraction in wound closure after brain injury in zebrafish larvae

Francois El-Daher[1,2], Stephen J Enos[2], Louisa K Drake[1], Daniel Wehner[2,3,4], Markus Westphal[2], Nicola J Porter[1], Catherina G Becker[1,2,5,*], Thomas Becker[1,2,*]

**Wound closure after brain injury is crucial for tissue restoration but remains poorly understood at the tissue level. We investigated this process using in vivo observations of larval zebrafish brain injury. Our findings show that wound closure occurs within the first 24 h through global tissue contraction, as evidenced by live-imaging and drug inhibition studies. Microglia accumulate at the wound site before closure, and computational models suggest that their physical traction could drive this process. Depleting microglia genetically or pharmacologically impairs tissue repair. At the cellular level, live imaging reveals centripetal deformation of astrocytic processes contacted by migrating microglia. Laser severing of these contacts causes rapid retraction of microglial processes and slower retraction of astrocytic processes, indicating tension. Disrupting the lcp1 gene, which encodes the F-actin–stabilising protein L-plastin, in microglia results in failed wound closure. These findings support a mechanical role of microglia in wound contraction and suggest that targeting microglial mechanics could offer new strategies for treating traumatic brain injury.**

## Introduction

Traumatic brain injury (TBI) is a severe condition affecting an increasing number of people, estimated from 27 to 69 million each year (Dewan et al, 2019; Guan et al, 2023). Brain trauma often leads to serious health problems including persistent impairments in cognitive, sensory, and motor function. Unfortunately, there is no treatment yet for this complex condition and brain tissue in mammals has a very limited capacity to repair. In contrast, zebrafish larvae possess high regenerative capacities and can fully repair brain injuries within a few days (Herzog et al, 2019). They are optically transparent, which allows for real-time in vivo visualisation of tissues and cellular processes. Moreover, these fish share a high degree of similarity in genetics, neuroanatomy, and social behaviour with humans. This makes them an attractive animal model for TBI-related studies (Zulazmi et al, 2021). In particular, little is known about the early cellular responses to TBI. One distinct feature of penetrating TBI is the formation of a wound. Understanding wound closure and the factors affecting its dynamics could bring important insights for therapeutic applications. Recovery of brain tissue integrity could be achieved by the insertion of new neurons and glial cells or by surviving neurons migrating to the injury site and filling the wound (Marz et al, 2011; Ibrahim et al, 2016). Microglia, the resident immune cells of the central nervous system (CNS), are the first cell type to respond to a brain injury and home in on the damaged tissue (Crilly et al, 2018; Herzog et al, 2019; Zou et al, 2021). They are recruited where the tissue is damaged, sequester chemicals, and engulf dead cells and cellular debris. Microglia are also known to interact with many cells, in particular neurons and astrocytes in the injured brain (Zambusi & Ninkovic, 2020). Interestingly, microglial cells can also exert physical forces (Bollmann et al, 2015) and could contribute to wound closure by pulling on the surrounding tissue. Therefore, these cells are a primary target for investigating their roles in brain injuries (Var & Byrd-Jacobs, 2020).

In a model of stab injury in the optic tectum, we show that such wounds close within 24 h. Tissue closes via large tissue deformations and not through migration or proliferation of neural cells. Live imaging and mathematical modelling suggest that mechanical forces from accumulating microglia could contribute to wound closure by pulling on astrocytic processes. Astrocyte/microglial contacts are under tension as shown by laser ablation. Furthermore, genetically and pharmacologically preventing microglial accumulation and destabilising F-actin in microglia, by disrupting lcp1, all inhibited tissue closure. These observations are all consistent with a mechanical role of microglia in wound healing as part of the inflammatory response after TBI. Overall, we propose a

---

[1]Centre for Discovery Brain Sciences, University of Edinburgh Medical School: Biomedical Sciences, Edinburgh, UK   [2]Center for Regenerative Therapies Dresden at the TU Dresden, Dresden, Germany   [3]Max Planck Institute for the Science of Light, Erlangen, Germany   [4]Max-Planck-Zentrum für Physik und Medizin, Erlangen, Germany   [5]Cluster of Excellence Physics of Life, TU Dresden, Dresden, Germany

Correspondence: francois.el-daher@ed.ac.uk
Francois El-Daher's present address is Institute of Quantitative Biology, Biochemistry and Biotechnology (IQB3), Centre for Engineering Biology, University of Edinburgh, Edinburgh, UK
*Catherina G Becker and Thomas Becker are equal senior authors

biophysical perspective on the early stages of brain tissue restoration after TBI, in which mechanical forces from microglia could have a pivotal role.

# Results

### Optic tectum wounds in injured brains close by tissue contraction

Investigating how wounds close after an injury is crucial to our understanding of how the brain can be efficiently repaired. We used an experimental paradigm of stab injury to the optic tectum in zebrafish larvae as previously described (Herzog et al, 2019). In this paradigm, the only immune cells in the brain parenchyma are microglia, identified by the expression of a transgenic reporter for p2ry12 and the expression of the 4C4 antigen (Herzog et al, 2019). The system thus offers the opportunity to observe the microglia's reaction to a brain wound in the absence of neutrophils, blood-derived macrophages, and cells of the adaptive immune system, which only develops later. To inflict the lesion, we inserted an insect pin at an angle into the middle of the optic tectum, avoiding peripheral proliferation zones and targeting areas where neurons have a laminar and columnar arrangement (Fig 1A).

The optic tectum is located in the upper part of the midbrain, which makes it an ideal target for penetrative injuries and live imaging. We took care when performing the direct lesion injury not to harm other parts of the brain, while still forming a clear wound in the periventricular zone (PVZ) initiating a microglial response.

To analyse the kinetics of wound closure, we injured 4 days post-fertilisation (dpf) Tg(h2a:GFP) zebrafish larvae in which cell nuclei were fluorescently labelled. We then performed 3-dimensional time-lapse imaging between 4 and 22 hours post-injury (hpi; Fig 1B and C; Video 1). We measured the GFP fluorescence intensity normalised between 0 (before wound closure) and 1 (total intensity in the wound at 22 hpi) over time in the median plane of time series inside the injured region of the PVZ. Here, the wound is most clearly delimited because of the compact arrangement of neuronal cell bodies (Fig S1A–D). Our data show that the stab wound is closed (i.e., filled by cells) between 18 and 22 hpi with similar kinetics among different animals (Fig 1D). Hence, robust wound closure is achieved in less than 24 hpi.

To investigate the mechanisms leading to wound closure, we first investigated whether cell proliferation can lead to the repopulation of the injury site. To detect newly generated neurons and other potential mitotic cells, we applied EdU from the time of injury in combination with immunofluorescence for neurons (HuC) and counted the newborn cells at 24 hpi. We detected labelled cells in the peripheral growth zones of the tectum, which served as an internal control for the EdU staining. However, in the injury site, there were no EdU or EdU/HuC double-labelled cells in five of six injured animals. In one fish, we observed 4 EdU/HuC double-labelled cells, and 6 EdU-only labelled cells in a superficial position that were probably displaced during the tissue preparation. We also did not observe any EdU+ cells in a corresponding position in three uninjured fish (Fig 1E). In time-lapse recordings of labelled neuronal cells in Tg(Xla.Tubb:DsRed), we also did not observe pre-existing neurons migrating to the injury site (Video 2) (Vandestadt et al, 2021). In some preparations, we noticed an increased number of HuC/D-positive cells in the neuropil region where the pin damaged the tissue. However, this likely represents extruded material from the periventricular area, caused by the removal of the pin during lesioning. In addition, Herzog et al showed, using the same injury method, that dead cells are present in the neuropil at 6 hpi (Herzog et al, 2019). Hence, wound closure in our experimental paradigm was not due to neurons filling in the injury site. Instead, whole-tissue movement may close the wound.

To characterise such movement, we followed individual neuronal cell nuclei in discrete locations in the ventricular cell layer by marking their nuclei using Tg(h2a:GFP) larvae in time-lapse recordings (Fig S2). These neurons are in a columnar arrangement and densely interwoven with astrocytic processes and can therefore be used to track tissue movement. After stab injury, tracked neurons globally showed a large displacement, whereas in intact animals, there was minimal movement (Fig 1F and G). Also, the trajectories of cells in the injured animals were relatively straight, compared with the small and random displacement of neurons in intact fish (Fig 1H). In injured fish, neurons located in the most caudal part of the tectum experienced a large displacement towards the neuropil. In contrast, those located on the rostral side showed a small lateral displacement towards the opposite hemisphere (Fig 1I). Kymograph analysis showed that neighbouring cells moved at similar speeds in parallel trajectories without overtaking each other (Fig 1J), which is expected for uniform tissue movement. Furthermore, we quantified in more detail cellular movements using 2D mean-squared displacement analysis (Dieterich et al, 2008) (Fig 1K). The analysis of injured zebrafish showed two phases: phase I indicates a superdiffusive behaviour with a power exponent of 1.86 (Fig 1L; 4–20 hpi), and phase II (20–22 hpi) indicates that the displacement slows down at longer times (>20 hpi), correlated with the plateau of the wound closure kinetic curve after 19 hpi (Fig 1D).

To further characterise putative tissue movements, we determined changes in the outline of the neuropil and ventricular border over time by performing time-lapse experiments where the animals were continuously imaged. We then analysed images from specific time points (typically first and last) from the time series. When required, images were registered for 3D motion as detailed in the Material and Methods section. In this part of the brain, the finely branched processes of specific cells, called ependymoradial glial cells or radial glia, are positive for the astrocyte marker glial fibrillary acidic protein and fulfil all astrocytic functions in homoeostasis and after injury (Jurisch-Yaksi et al, 2020). For that reason, we call these processes astrocytic processes.

For visualising the neuropil that comprises both neurites and astrocytic processes, we used Tg(her4.3:GFP-F); Tg(elavl3:MA-mKate2) double-transgenic animals, labelling the dense network of astrocytic processes and neuronal membranes with the membrane-tethered fluorophores GFP and mKate2, respectively (Fig 2A).

Indeed, astrocytic processes support the structural integrity of the optic tectum and should be involved in any whole-tissue movement or deformation (Grupp et al, 2010; Arenzana et al, 2011). We observed that the neuropil underwent deformation,

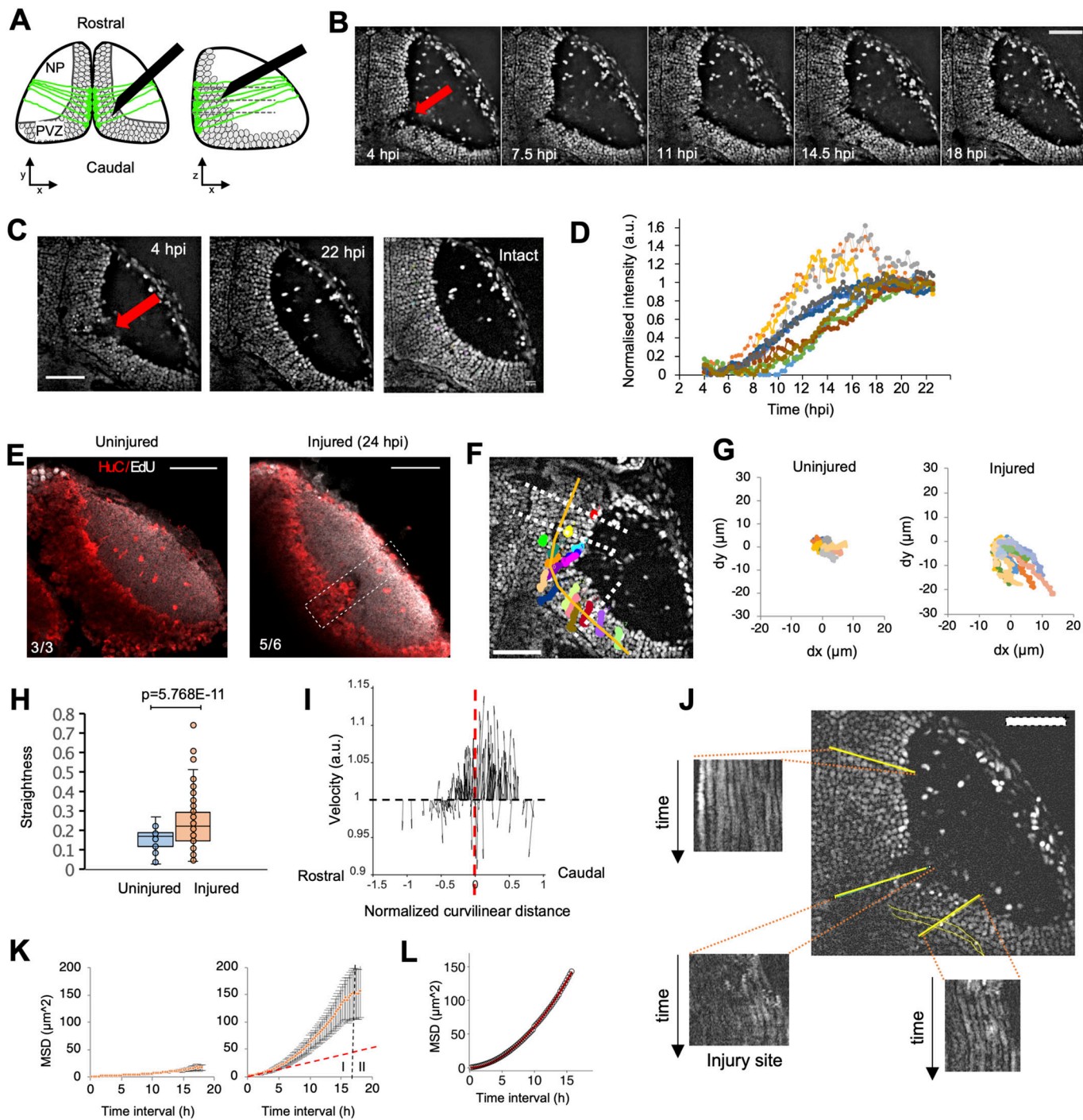

**Figure 1. Optic tectum wound closes within 24 h after the injury.**

**(A)** Organisation of neurons in the optic tectum and our injury model. Neurons (dark circles), surrounded by astrocytic processes (green), are arranged in layers (lamina, dashed lines) and columns as shown in XY and XZ sections; an insect pin (black) is used to injure the optic tectum down to the periventricular zone. NP: neuropil, PVZ: periventricular zone. **(B)** Time series of Tg(h2a:GFP) 4 dpf larvae after an injury showing wound closure (red arrow indicates wound position). **(C)** Wound closure between 4 hpi (left) and 22 hpi (middle) shows the restoration of the tissue as compared to the intact tectum (right; red arrow indicates wound position). **(D)** Variation over time of the fluorescence intensity inside the injury site during tectum repair (n = 11 larvae). **(E)** EdU staining (white) and HuC immunofluorescence (red) in uninjured Tg(h2a:GFP) larvae (left) and injured Tg(h2a:GFP) larvae (right). **(F)** Individual trajectories of neuron nuclei in the tectum of injured Tg(h2a:GFP) 4 dpf larvae (coloured traces). The starting points of individual trajectories were projected perpendicularly (white lines) on the curved rostrocaudal extend (orange curve). **(G)** Individual trajectories in uninjured and injured larvae showed an increased and directional XY displacement for injured animals. **(H)** Straightness analysis of uninjured (n = 7) and injured (n = 11) larvae showing that trajectories of neuronal nuclei are more elongated after injury. Box plots show the median, box edges represent the 25th and 75th percentiles, and whiskers indicate ± 1.5 x the interquartile range. ($P$ <0.0001, $t$ test). **(I)** Trajectory directionality analysis of individual nuclei after rostrocaudal curve linearisation indicates trajectory anisotropy. The injury centre position is represented by the dashed red line. **(J)** Kymographs at different locations in the PVZ area of an injured fish

with a contraction of the caudal part of the tectum and a dilation perpendicular to the curved rostrocaudal extent in the rostral part of the tectum (Fig 2B and C). We measured the ventricular border deformation using Tg(h2a:GFP) larvae (Fig 2D). The rostral part showed a deformation towards the contralateral side, and the caudal part showed a deformation towards the rostral part (Fig 2E). This is consistent with the deformations observed for the neuropil and with the concerted trajectories of individual cells described above. Hence, the injured optic tectum tissue appears to be subject to anisotropic deformations during wound closure.

### Wound closure is due to mechanical forces that induce tissue deformations

Tissue deformation during wound closure could result from the passive movement of the tissue into the gap or from active forces pulling on the tissue. To investigate whether the observed kinetics of wound closure could be explained by active forces, we developed a theoretical model that assumes elastic forces acting on the tissue. This model assumes that neurons are bound to a spring in a viscous medium (Fig 2F) and follow the dynamics of a viscoelastic harmonic oscillator. The spring represents the elasticity of axons bundled to each other by protocadherins (Cooper et al, 2015) and of astrocytic processes embedding neuron cell bodies (localised in the PVZ as shown by images of their nuclei) and axons. We considered only one half-period of oscillations as we observed the displacement to be monotonous towards the neuropil. We generated theoretical displacement curves of neurons, and considering them as landmarks of the PVZ tissue, we compared them with the experimental wound closure kinetic curves. The theoretical curves obtained from the model were in close agreement with the experimental kinetic curves (Fig 2G). This is consistent with an external elastic force driving neural tissue displacement.

To experimentally test whether active forces could be involved in tectum repair, we used blebbistatin, a drug that inhibits non-muscular myosin II activity and thus reduces force generation by cells relying on the actomyosin complex (Várkuti et al, 2016). We first performed brain injuries using the same experimental conditions and then treated injured larvae with 1 $\mu$M blebbistatin from 7 to 24 hpi. This delayed drug application allowed microglia to migrate to the injury site (Herzog et al, 2019). To quantify the efficiency of wound closure and the effect of blebbistatin, we defined a repair index as the ratio of the injury volumes at 4 and 24 hpi. We used this approach on 9 control and 12 treated fish and found that wound closure was significantly impaired in treated larvae (Fig 2H and I).

Altogether, these analyses suggest that after injury of the optic tectum, the wound closes via tissue deformation caused by mechanical forces involving the actomyosin complex.

### Mechanical forces originate from a region where microglia accumulate

We next investigated where the mechanical forces responsible for the wound closure originated. Using cell nuclei as landmarks of the tectal PVZ to characterise local tissue deformations, we performed a displacement field analysis on data presented in Fig 1. This showed that trajectories of nuclei converged towards the neuropil at an average normalised tectum radial position of 0.6, with an average angular position at 37° (Fig 3A and B). This indicated a relatively narrow singular origin of potential contraction forces in the superficial neuropil centred over the injury site.

To characterise the origin site of the contraction force in vivo, we re-examined images of injured Tg(h2a:GFP) larvae. We identified an accumulation of cell nuclei near the convergence point of neuronal trajectories in the neuropil in all animals (n = 11; Fig 3C). These cells could be microglia, based on previous observations that these cells are recruited in the neuropil of zebrafish larvae in the first 2 h after injury (Herzog et al, 2019). Using Tg(mpeg1:mCherry) transgenic larvae, we observed that the accumulated cells were labelled by the transgene mpeg1, a marker of macrophages and microglia (Fig 3D). We further identified the cells as microglia using immunofluorescence with the 4C4 antibody, which specifically recognises microglia (Becker & Becker, 2001; Rovira et al, 2023) (Fig 3E). To confirm that the accumulation site of microglial cells corresponds to the potential centre of force generation, we compared the position of the convergence points of the trajectories of neuronal cell nuclei, which we took as landmarks of the direction of tissue movement, with the position of microglial cells. We analysed the position of 308 individual microglial cells from 15 Tg(mpeg1:mCherry) larvae at 8 hpi, a time point when tissue movement after injury was fast (Figs 3F and S3). We determined the accumulation centre to be in the neuropil at a normalised radial position of 0.88 and an average angular position of 17°. This is close to the centre of convergence of the neuronal trajectories determined earlier and is consistent with microglia being at the origin of the tissue pulling forces.

Next, we determined the temporal correlation between microglial accumulation and wound closure. We first performed time-lapse imaging on injured Tg(Xla.Tubb:DsRed); Tg(mpeg1:GFP) double-transgenic larvae (Fig 3G, Video 2), in which neuronal cell bodies and microglia were fluorescently labelled in the same animal. This showed that microglia were recruited from 2 hpi (first time point of observation), rapidly accumulated until 6 hpi, and stayed accumulated in the neuropil for at least 12 hpi (Fig 3G). Additional observations in histological preparations showed that microglia remained accumulated for at least 24 hpi (data not shown). Without injury, microglia did not accumulate and were rarely present in the neuropil itself in contralateral hemispheres or uninjured brains (Fig 3H). Microglia move mainly in the PVZ between neuronal somata in these unlesioned situations. By comparing the

---

show that neurons keep their laminar organisation over time. **(K)** Mean-squared displacement (MSD) analysis of individual trajectories of neuronal cell nuclei for uninjured (left) and injured (right) larvae shows two phases of displacement: I, a superdiffusive behaviour after injury; and II, a slower increase compatible with diffusion. The red dashed line shows the MSD for a diffusive case. **(D, L)** MSD superdiffusion model, MSD(t) Ã t—fit of the experimental MSD curve in (D). Blue curve: theoretical model; red points: experimental values. R > 0.999, — = 1.86. All images are oriented with the rostral side up. Scale bars represent 50 $\mu$m on all images.

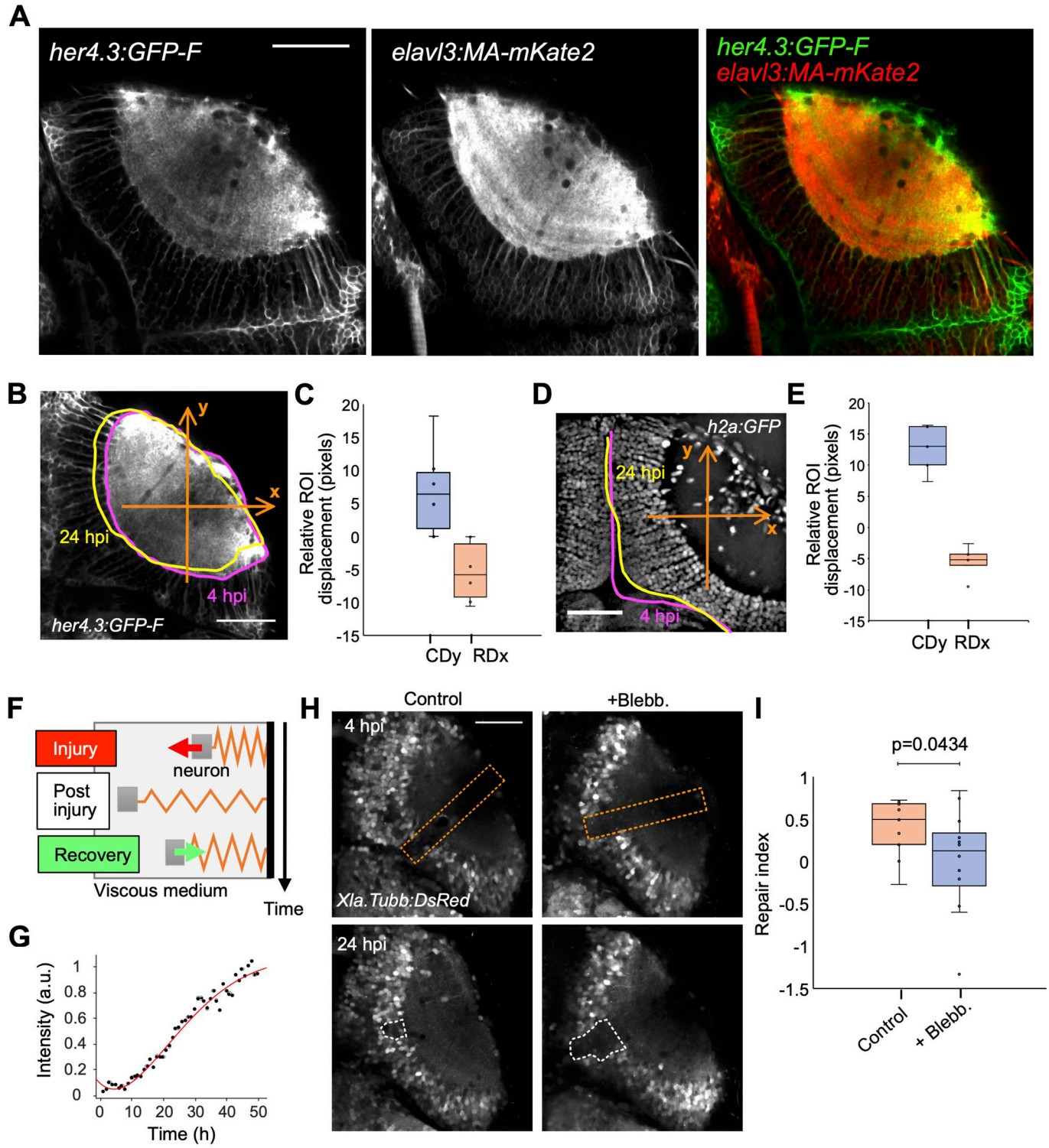

**Figure 2. Optic tectum wounds close via tissue deformation.**
**(A)** Fluorescence images of a Tg(her4.3:GFP-F); Tg(elavl3:MA-mKate2) fish at 4 dpf. **(B)** Measurement of the neuropil contour deformation using Tg(her4.3:GFP-F); Tg(elavl3:MA-mKate2) larvae. Deformation is measured along the axes shown (orange arrows). **(B, C)** Boxplot representation of the neuropil contour deformation measured in (B) (n = 6), represented on the caudal (y-axis, CDy, blue) and rostral (x-axis, RDx, orange) axes. **(D)** Deformation quantification using the PVZ boundary in Tg(h2a:GFP) larvae. **(D, E)** Boxplot representation of the deformation measurements as performed in (D) (n = 6). **(F)** Model of viscoelastic dynamics of neuron displacement comprising three phases (see Supplemental Information). **(G)** Example of experimental wound closure curve (black dots) fitted by the model (red curve). **(H)** Example images of injured optic tectum of control and blebbistatin-treated Tg(Xla.Tubb:DsRed) animals at 7 and 24 hpi. The orange dashed lines show the position of the injury. The white dashed lines show the analysis regions for calculating the repair index. **(I)** Quantification of the repair index (RI) for control (n = 9) and blebbistatin-

kinetics of wound closure and microglial accumulation in the neuropil, we determined that the wound starts to close only after microglia have fully aggregated in the neuropil at 6 hpi (Fig 3I). This is consistent with microglia as the origin of contraction forces.

To analyse the spatiotemporal correlation between microglial and tissue local deformations in more detail, we generated additional time-lapse movies of Tg(Xla.Tubb:DsRed); Tg(mpeg1:GFP) larvae at a higher frame rate to track microglia and neuronal cell bodies at the same time. We tracked neuronal cell bodies as tissue landmarks and microglia to compare their trajectories after injury. We observed that during the microglial accumulation phase (after 6 hpi), the tissue was deformed (measured by tracking neuron nuclei) towards the neuropil. Local deformations appeared to follow the movement of microglia compacting at the injury site (Fig 3J). To estimate the similitude in terms of shape between trajectories, we computed the discrete Fréchet distance between microglial trajectories and the experimentally observed neuronal trajectories. To estimate whether the value represents a good match between the movements, we compared it with that for randomised neuronal trajectories (Fig 3K). The much higher Fréchet distance when using randomised trajectories indicated that local tissue deformation dynamics correlated well with microglial movement.

Overall, these results show that microglial accumulation is at the origin of the elastic forces inducing tissue deformation and, therefore, wound contraction.

### Computational modelling indicates sufficiency of potential microglial pulling behaviour for wound repair

During their migration, microglial cells likely exert forces on their surroundings (Steinwachs et al, 2016). To estimate whether forces exerted by microglia could be sufficient to explain wound closure, we first took a theoretical approach by building a minimal system mimicking the brain tissue and microglial activity. To this aim, we developed a multi-agent system using the PhysiCell framework (Fig 4A and Supplemental Information) using three agents: neurons, microglia, and skin cells. In the absence of external forces, neurons were allowed only to have a small random displacement as observed experimentally in intact fish. Skin cells were immobile and introduced to mimic the external tissue boundaries and stiffness.

Microglia were allowed to migrate and exert elastic forces on all the other agents, following our first model of elastic forces controlling the displacement of neurons (see Fig 3F). We then used this system to perform in silico experiments with an initial microglial distribution as isolated cells spread in the neuropil. We could reproduce both microglial accumulation and wound closure because of tissue contraction (Fig 4B, Video 3). Moreover, the kinetics of wound closure measured by the intensity of neuronal cells in the injury site over time (similar to the method used for experimental data) presented a curve that was a good fit for the experimentally observed wound closure curve (Fig 4C). When microglial cells were reduced in number or even absent in the simulation (Fig 4D and E), we found a linear relationship between wound closure, expressed

as a repair index, and microglial number until a plateau is reached when the number of microglia is sufficient to induce a full closure. In the extreme case where no microglia were present, the wound did not close. Therefore, our model predicts that the repair process depends on the number of microglia accumulating and that potential mechanical forces exerted by these cells could lead to brain tissue contraction.

### Microglia are essential to restore the optic tectum architecture after injury

To experimentally test the model's predictions, we first depleted microglia genetically, using the irf8 mutant (Shiau et al, 2015), and pharmacologically, using the Csf1r inhibitor KI20227 (Chia et al, 2018). Irf8 is a transcription factor that is essential for the development of all macrophages, including microglia. Irf8 mutants are adult-viable but selectively lack microglia and macrophages (Shiau et al, 2015). To visualise the tectal tissue, the larvae also carried the Tg(Xla.Tubb:DsRed) transgene, which labels the neuronal tissue. We observed that in 16 of 21 WT control animals, the wound was closed at 24 hpi, as before (R.I. > 0.5). In contrast, the wound was closed in only 2 of 17 irf8 mutants (Fig 4F).

We then compared the repair index between WT and irf8 mutants at 24 hpi (Fig 4G), indicating a significantly lower repair index in the mutant. This was independent of the initial injury volume because there was no correlation between the initial injury volume and the repair index across all animals (Fig 4H). We then measured the trajectories of neuronal cell bodies in irf8 mutants to assess the tissue deformation. We found that trajectories were directed more often towards the outside of the tectum, with a larger amplitude for those oriented opposite to the neuropil, compared with injured WT larvae (Fig 4I and J). Furthermore, from these measurements, we determined that the injury volume increases in Tg(Xla.Tubb:DsRed); Tg(irf8−/−) larvae in 59% (10 of 17 animals) of the cases and never in WT larvae. In these mutant animals, the wound enlarged, and the tectum dilated instead of contracting. Similarly, the KI20227 treatment, which markedly depleted microglia (Fig 4K), reduced the repair index in treated animals (Fig 4L). Hence, the results of these manipulations align with the model's prediction and indicate that microglia are necessary for wound closure.

### Microglia interact with astrocytic processes

To directly observe how microglia physically interact with the surrounding tissue during accumulation, we focused on astrocytic processes because these are finely branched and widely distributed throughout the tectum. To visualise these astrocytic processes in fine detail, we used our newly generated Tg(her4.3:GFP-F) line. We performed 3D time-lapse image acquisitions on injured Tg(her4.3:GFP-F); Tg(mpeg1:mCherry) double-transgenic larvae, in which astrocytic processes Tg(her4.3:GFP-F) and microglial cells Tg(mpeg1:mCherry) were fluorescently labelled in the same animals, during a 4- to 24-hpi time window.

treated (n = 12) fish. t test, P < 0.05. Box plots show the median, box edges represent the 25th and 75th percentiles, and whiskers show the full data range. Scale bars represent 50 µm on all images. All quantifications are between 4 and 24 hpi unless otherwise stated.

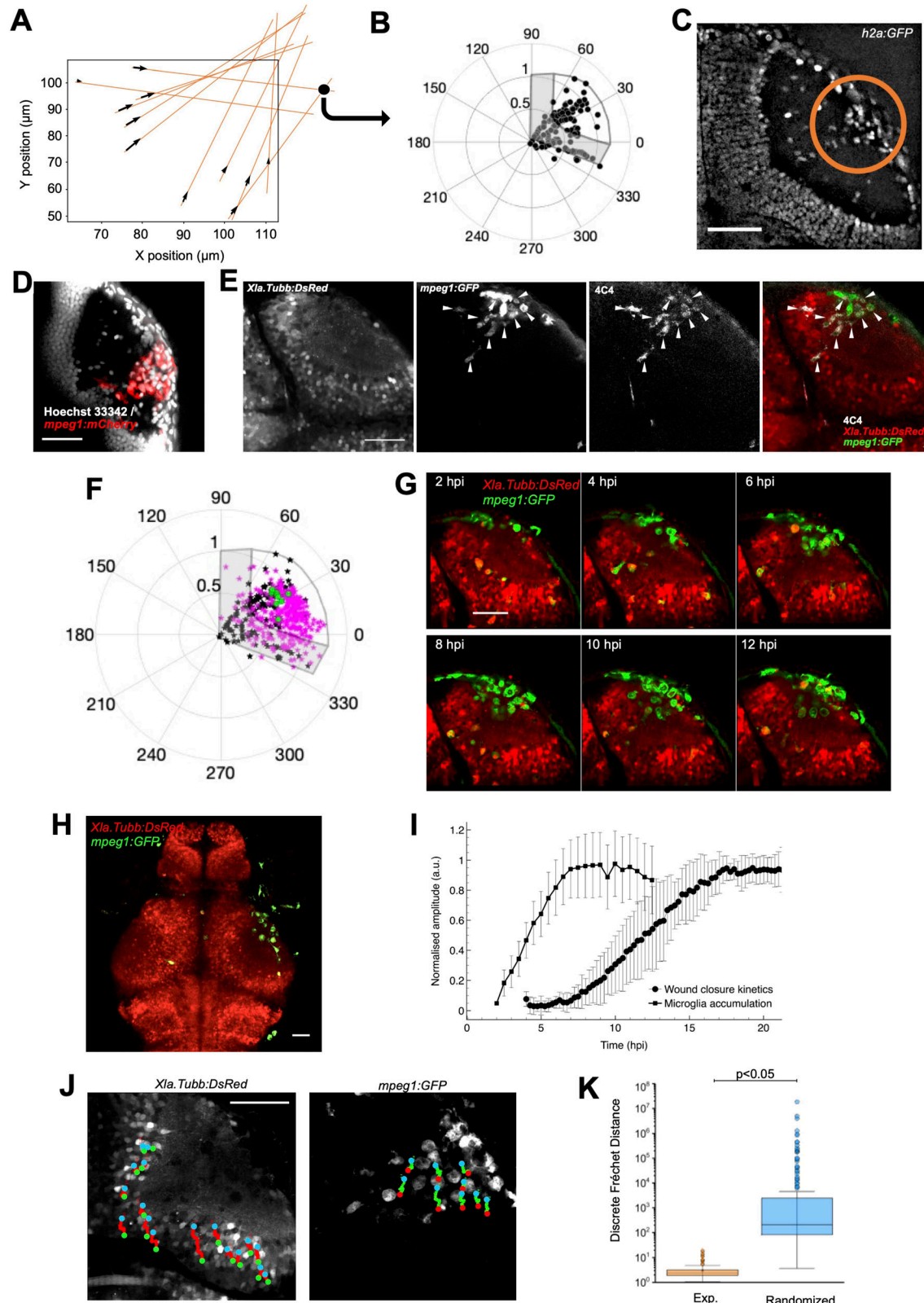

**Figure 3. Microglial invasion in the neuropil correlates with the kinetics of wound closure.**
**(A)** Displacement field analysis showing the direction of displacement of individual neuronal nuclei (black arrows). The orange lines show how the intersection points (black spots) between trajectories are estimated. **(B)** Polar representation (in degrees) of the position of intersection points of individual trajectories of neuronal nuclei from 10 animals after tectum size normalisation and rotation. The optic tectum is represented as an overlay (boundaries in dark grey and PVZ in light grey).

We observed an increase in the fluorescence signal for astrocytic processes in maximum intensity projections around the microglial accumulation site (Fig S6C, Video 4), which may indicate interactions between astrocytic processes and microglial cells. We then assessed at high magnification whether the alteration of the meshwork of astrocytic processes was related to microglial dynamics by performing fast time-lapse imaging on injured Tg(her4.3:GFP-F); Tg(mpeg1:mCherry) larvae, starting at 6 hpi when microglia were stably present. We observed astrocyte and microglial interactions (Fig 5A, Video 5) that seem to typically occur in three consecutive steps that we termed SAT: S—sensing; A—adhesion; and T—traction. In phase I (sensing), microglia generate transient protrusions in apparently random directions and occasionally form longer lasting associations with astrocytes shown by the colocalisation of both signals (phase II, adhesion). During phase III (traction), microglia retract their protrusion leading to the deformation of astrocytic processes following the retracting microglial process. We observed more generally that astrocytic processes seemed to be pulled by microglia when microglia moved ("migration" events) inducing additional deformation of the meshwork (Fig 5A and B; Video 6). More surprisingly, we observed an original phenomenon that we named "astrocytic knitting" where astrocytic processes seem to be "glued" together after the traction phase (Fig 5C, Video 7).

Overall, from the series of 30- to 40-min time-lapses acquired at 6 hpi, we observed 54 SAT events from the analysis of 43 microglial cells accumulated in the neuropil (from three animals) and estimated an average frequency of 20 SAT events/hour and per hemitectum. The frequent occurrence of the SAT processes during the repair phase shows that microglia transiently attach to astroglial processes during their migration, leading to their deformation. This interaction could lead to tissue contraction.

These observations align with the notion that microglia may exert a mechanical action on the tissue, for example, by pulling on astrocytic processes.

### Micro laser ablation of microglial–astrocytic contacts leads to retraction of processes

To obtain direct evidence for a potential microglial pulling action on astrocytic processes, we targeted these contacts in WT animals with a 2-photon laser and imaged a 3D region of interest (ROI) within the injury site in Tg(her4.3:GFP-F); Tg(mpeg1:mCherry) double-transgenic larvae at 16–18 hpi at every 15 s for 5 min. We measured the change in the area of microglia and astrocytic processes. For the microglia, we observed a large retraction of the mostly elongated contact site within the first frame (6/6 cases; Figs 6A and B and S4A; Video 8).

Quantifying the area change of the microglia indicated a change in the first 15 s after injury that was 30-fold larger than in any subsequent frame, and more than threefold greater than in microglia migrating towards a laser injury site from a distance (Figs 6C and S4B). Of note, microglia with severed processes resumed movement after 30–45 s (six of six cases) and, hence, were not destroyed by the laser.

Because we observed impairment of wound closure in animals treated with blebbistatin (Fig 2I), we repeated the experiments using this treatment. As expected, the initial area change was strongly reduced compared with experiments in which microglial–astrocytic contacts were severed in the absence of blebbistatin (3/3 cases; Figs 6C and S4C). To achieve higher temporal resolution, we recorded only one optical section with a time resolution of 0.65 s. We managed to observe two thin microglial processes that retracted almost immediately. Within the first 3 s after severing the contact (Fig S5, Video 9), they moved at an average speed of 52 $\mu$m/min (0.89 $\mu$m/s). After that, processes move towards the injury site again at a much lower speed (0.18 $\mu$m/s), indicating survival of the cells and migration towards the injury. Hence, also at higher temporal resolution, recoil speed greatly exceeds the migration speed of microglia. This indicates that microglial processes after laser severance quickly retract, consistent with the contact being under tension.

Astrocytic processes showed retraction in the opposite direction to the microglia (Fig 6A–E). However, within the first minute after severing the contact, we did not observe significant movement of the astrocytic processes (Fig 6A and C).

Analysing the change in the area for the astrocytic processes for a longer time period indicated that their reaction was more protracted and cumulative area change became significant only at 4 min after cutting the contact with microglia (Fig 6B and D). Injuring astrocytic processes in places without microglia in the vicinity (>5 $\mu$m) did not lead to a displacement, indicating dependency of the movement on the previous microglial contact (Fig S4D). Hence, astrocytic processes show a specific, but slow, reaction to the severance of microglial contacts. In summary, the retraction of astrocytic processes and very rapid retraction of microglial processes from a severed contact point between these cell types are consistent with a potential mechanical interaction between astrocytic and microglial processes (Fig 6E).

---

(C) Accumulation of cells in the neuropil (inside the orange circle) after injury in Tg(h2a:GFP) larvae at 20 hpi. (D) Identification of the accumulated cells as mpeg1+ cells using Tg(mpeg1:mCherry) (red) larvae, combined with nuclear staining (Hoechst 33342, grey). (E) Identification of the mpeg1:GFP+ cells as microglia using 4C4 immunofluorescence in an injured Tg(Xla.Tubb:DsRed); Tg(mpeg1:GFP) fish. White arrows point out mpeg1:GFP+/4C4+ double-labelled cells. (F) Polar representation of microglial positions (N = 308 from 15 animals, magenta) in the neuropil compared with intersection points of trajectories of neuronal nuclei (black, green spots: centre of gravity of intersection points for each animal). (G) Selected frames of a time-lapse imaging series of an injured Tg(Xla.Tubb:DsRed); Tg(mpeg1:GFP) transgenic animal show the recruitment and accumulation of microglia in the injury site. (H) Lower magnification image of an injured Tg(Xla.Tubb:DsRed); Tg(mpeg1:GFP) transgenic animal showing that microglia did not accumulate in the intact contralateral tectum. (I) Kinetics of microglial accumulation from 2 hpi to 13 hpi (black square) and the average wound closure kinetics (black dots). (J) Dual-channel single-cell tracking of neuronal cell bodies (left, red trajectories) and microglia (right, green trajectories) in an injured Tg(Xla.Tubb:DsRed); Tg(mpeg1:GFP) larva. (K) Boxplot representation of the discrete Fréchet distance estimation between microglial trajectories and experimental (orange) and randomised (blue) trajectories of neuronal cell nuclei. Box plots show the median, box edges represent the 25th and 75th percentiles, and whiskers show ± 1.5 x the interquartile range. t test, P <0.05. Scale bars represent 50 $\mu$m on all images.

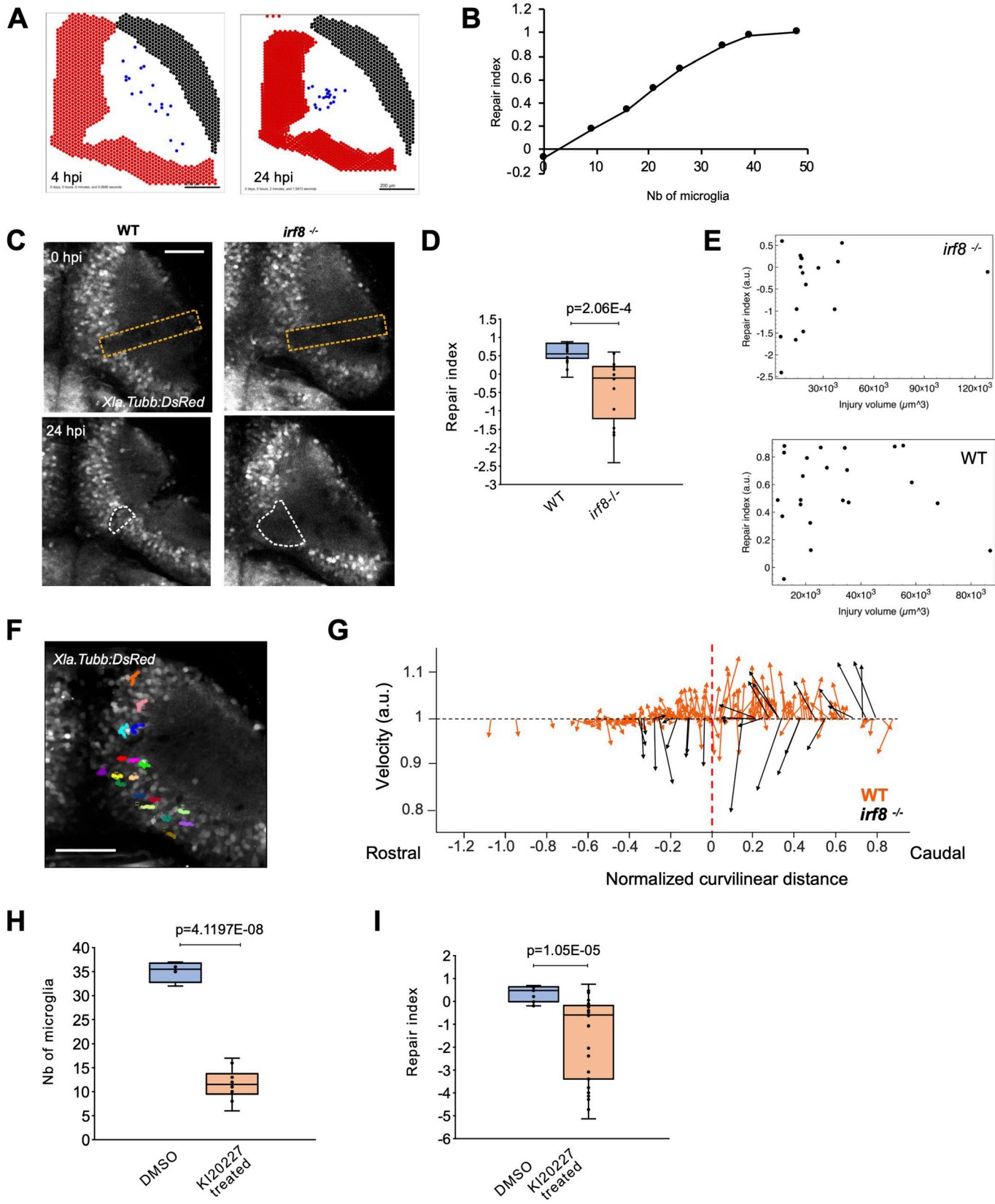

### Microglial action on wound closure depends on lcp1 gene function

If microglial cells were contributing to tissue deformation by exerting local traction forces on the surrounding tissue, they need to contain non-muscular myosin II in addition to actin. We confirm this here by immunofluorescence against non-muscular myosin IIB/myh10 in Tg(mpeg1:GFP) animals, showing enrichment in tectal microglial cells (Fig S6B). To detect the actin cytoskeleton, we used Tg(mpeg1:GFP); Tg(beta-actin:utrophin-mCherry) animals. These are double-transgenic animals, in which the actin-binding protein utrophin is labelled with mCherry and microglial cells in the tectum are labelled by GFP. In these animals, we observed strong labelling of the microglial cell cortex with mCherry, compared with the cell's centre and the surrounding tissue in the neuropil (Fig S6A). Moreover, we observed a close association of the actin cytoskeleton in microglia with SAT events (Fig S7; Video 10). These observations indicate the presence of the cellular machinery for force generation in microglia, correlated with a potential contribution to tissue deformation.

To determine whether the magnitude of a potential microglial traction force could be relevant to wound closure mechanics, we decided to weaken mechanical actions in a cell type–specific manner in microglia. To achieve this, we decided to ablate L-plastin function in microglia. L-plastin is coded for by the gene *lcp1*, which is specifically expressed in immune cells under physiological conditions. In our tectum lesion assay, only microglia are present (see 4C4 immunofluorescence in Fig 3E and Herzog et al [2019]). L-plastin stabilises bundles of filamentous actin (F-actin) and is thus essential for force generation, for example, in cancer cell invadopodia (Van Audenhove et al, 2016). Importantly, F-actin is also highly enriched in protrusions of brain macrophages that exert traction forces on endothelial cells during blood vessel repair (Liu et al, 2016), suggesting that lack of L-plastin weakens F-actin function in traction events.

First, we confirmed the expression of lcp1 in microglia. To do so, we performed double HCR for the microglial marker apoeb (Mazzolini et al, 2020) and lcp1 and observed an accumulation of lcp1-expressing cells in the injured tectum at 6 hpi (Fig 7A). Of these cells, 92 ± 3.69% were apoeb-positive, indicating the nearly exclusive expression of lcp1 in microglial cells in the injured tectum (Fig 7B). Hence, any manipulation of *lcp1* will be specific to microglial cells.

To functionally analyse the relevance of lcp1 in our wound closure assay, we designed highly active sCrRNAs to effectively create somatic mutants (Keatinge et al, 2021). Restriction fragment length polymorphism (RFLP) analysis after zygote injection confirmed that highly active sCrRNAs were indeed highly efficient in vivo in disrupting a targeted restriction enzyme recognition sequence (Fig S8). Somatic mutation of lcp1 did not overtly impair the accumulation of microglia in the injury site, and no differences in the number of microglial cells were observed (Fig 7C), indicating that microglia could still perform potential functions that were unrelated to force generation. However, in lcp1 somatic mutants, the tectum wound failed to close completely. Quantification of all samples, or when a cut-off for microglial presence in the tectum was set (>50 cells), indicated that the repair index was significantly lower than in controls in both analyses (Fig 7D and E). This indicates that the F-actin–bundling L-plastin protein in microglia is necessary for effective closure of the tectum wound. In summary, these observations provide evidence for potential mechanical forces generated by microglia that are strong enough to lead to brain wound contraction.

## Discussion

Here, we present evidence that microglia could play a previously unknown role in the CNS tissue repair by facilitating contraction of the injured tissue, leading to the closure of a physical wound.

Direct evidence for the potential role of microglia in pulling on the surrounding tissue comes from our observation that microglia displace astrocytic processes and that severing microglial–astrocyte contacts leads to a rapid recoil of microglia and a protracted pull-back of astrocytes. This suggests physical tension between these cellular elements. At high temporal resolution, we observed an almost immediate withdrawal of microglial processes at speeds that were consistent with other paradigms testing tissue tension (Stachowiak & O'Shaughnessy, 2009; Mao et al, 2013; Ventura et al, 2022; Baraban et al, 2023). The reasons for the slower, but consistent, movement of the astrocytic processes need further investigation but may lie in their meshwork-like arrangement, rather than being isolated entities, like the microglial cells. Our observations are supported by other studies showing that microglia can exert traction forces on their substrate and sense differences in tissue stiffness (Bollmann et al, 2015; Ayata & Schaefer, 2020). In addition, the role of macrophages in blood vessel repair with direct mechanical action at the cellular scale has been demonstrated in vivo (Liu et al, 2016). Recently, a study has demonstrated the importance of microglia for the maintenance of the mechanical

---

**Figure 4. Microglia are necessary for brain tissue repair.**
**(A)** Example of simulation result showing microglia accumulation and wound closure. Left: simulated tissue after injury with patches of microglia agents in the neuropil. Right: the same simulation at 24 hpi. **(B)** Relation between the repair index estimated on the simulated data and the number of microglia agents in the simulation. **(C)** Images of an injured tectum in WT and irf8−/− Tg(Xla.Tubb:DsRed) larvae, right after the injury (0 hpi) and at 24 hpi. The orange dashed lines show where the injury was made. The white dashed lines show the analysis regions for calculating the repair index. Scale bar: 20 μm. **(D)** Quantification of the repair index (=1 − V@24 hpi/V@4 hpi) in WT (n = 21) and ifr8 mutant larvae (n = 17). *t* test *P*-value <0.001. **(E)** Correlation analysis of the repair index versus initial injury volume shows that the repair index is independent of the initial size of the injury. **(F)** Trajectories of neuronal cell bodies in an injured irf8 mutants (Scale bar: 50 μm). **(G)** Comparison of anisotropy of trajectories in WT (orange, from Fig 1I) and in ifr8 mutants (black). **(H)** Quantification of the depletion of microglia using KI20227 *t* test *P*-value <0.0001 (Controls n = 4, treated n = 10). **(I)** Measurement of the repair index in control (n = 12) and KI20227 treated (n = 26) animals. *t* test *P*-value <0.0001. Box plots show the median, box edges represent the 25th and 75th percentiles, and whiskers show the full data range. Scale bars represent 50 μm on all images.

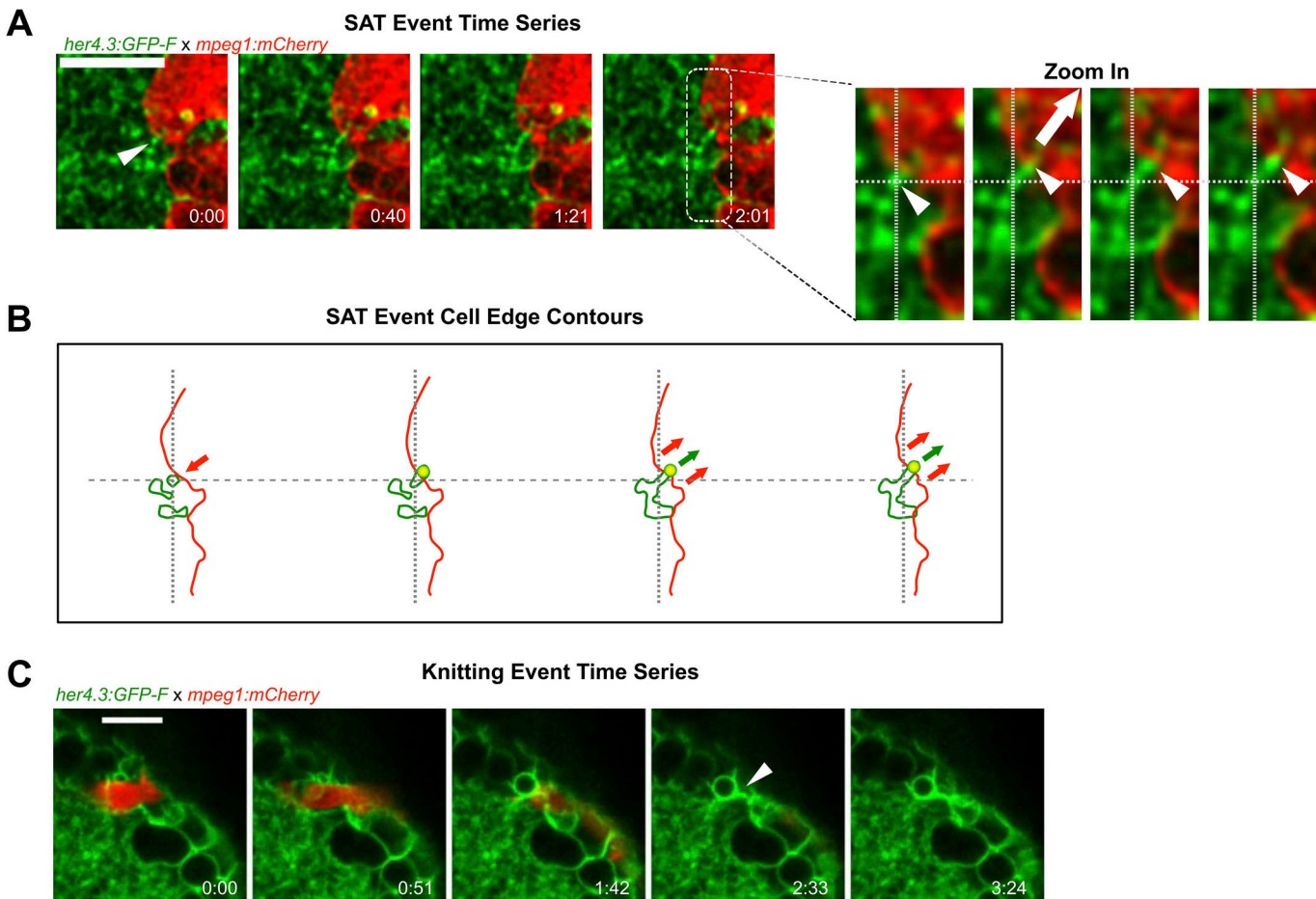

**Figure 5. Microglial contacts displace astrocytic processes.**
**(A)** Fast time-lapse imaging sequence on a Tg(her4.3:GFP-F);Tg(mpeg1:mCherry) larva showing an SAT process. The white arrow points out the adhesion event. (A') Cropped sequence from (A). The small white arrows point to astrocytic protrusion pulled by microglia protrusion. The big arrow shows the direction of the traction force. The dashed line indicates the initial and final positions. **(A, B)** Outline of microglia (red) and astrocytic structures (green) from images in (A). Arrows show the direction of displacement. **(C)** Example of a knitting event. All scale bars: 10.

properties of the brain tissue in zebrafish larvae (So et al, 2024 Preprint).

We observed directed tissue movements and microglial accumulation correlated with wound closure. This prompted a modelling-based approach that predicted that mechanical forces by microglia could be sufficient for wound closure. Our mathematical model is highly simplified, for example, assuming equal forces between microglia and other cellular elements. Nevertheless, even with these minimal assumptions, the model accurately predicted impaired wound contraction in the absence of microglia in irf8 mutants and reduced abundance after Csf1ra inhibition and thus suggests that microglial pulling action could theoretically be sufficient to close a brain wound in our model. However, the absence of microglia may also impact the outcome of injuries indirectly. For example, the lack of cytokine signalling to astrocytes (Var & Byrd-Jacobs, 2020) or other cell types is likely to contribute to impaired wound closure. In spinal cord regeneration, macrophage-derived cytokines are crucial (Tsarouchas et al, 2018). Nevertheless, in experiments targeting cytoskeleton function, that is, blebbistatin administration, and

in lcp1-deficient animals, microglia were still present in the tectum, suggesting that at least some of their signalling functions were retained. Therefore, these experiments better align with the hypothesis that microglia have a role in tissue force generation that is essential in tissue wound closure.

Regarding the cellular machinery involved in the pulling action of microglia, we find that microglia-specific lcp1 is necessary for efficient wound closure of the tectum. The protein product L-plastin stabilises F-actin bundles that are critical for force generation in stress fibres (Van Audenhove et al, 2016). Lcp1-deficient microglia are the only lcp1-expressing cell type in the injured tectum. Hence, the loss of lcp1 in microglia in lcp1 somatic mutants specifically weakens F-actin–dependent force generation in microglia. Ablation of lcp1 function thus is in agreement with the idea that tension between astrocytic processes and microglia may be sufficiently large to make a meaningful contribution to wound closure at the tissue level. Interestingly, lcp1-deficient macrophages undergo shape changes after tail injury in larval zebrafish, consistent with an inability to exert pulling forces (Linehan et al, 2022).

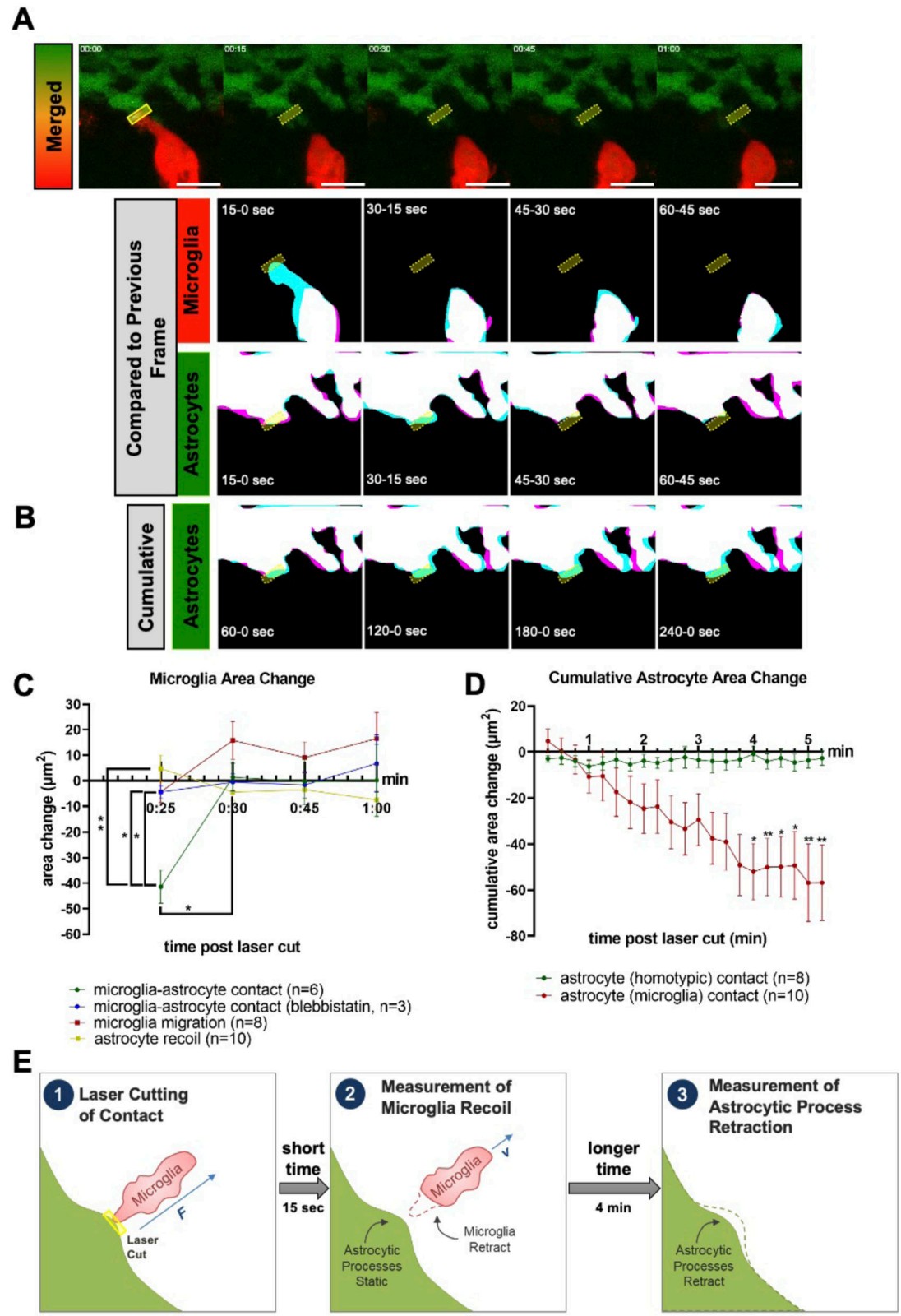

**Figure 6. Microglia and astrocytic process recoil after laser severance of contact sites.**
**(A)** Diagram of laser-induced microglial recoil away from astrocytic contacts and slower retraction of astrocytic processes. **(B)** Time-lapse images of microglia (red) and astrocytes (green) after laser ablation (upper, up to 1 min post-ablation), with area masks overlaid between adjacent time points for microglia (middle) and astrocytes (lower). **(C)** Cumulative area masks of astrocytes for 1–4 min post-ablation. **(B, D)** Change in area after laser ablation for microglia that were contacting astrocytes (green,

Failure to close the wound in the absence of microglia could also be explained by the lack of debris removal, another microglial function (Herzog et al, 2019). However, in injured irf8 mutants, neutrophils invade the tectum, which they did not do in WT animals, and were able to clear dead cells when microglia were missing (Zou et al, 2021). This suggests that the inability to close a wound, at least in the irf8 mutant, is unlikely to be a consequence of impaired removal of dead cells.

The role of microglia in human TBI is not fully understood. Increased microglial inflammation leads to secondary damage (Cole et al, 2018, Todd et al, 2021), but may also have positive functions for recovery (Lalancette-Hebert et al, 2007). Human microglia and zebrafish telencephalic microglia show similar gene expression signatures after injury (Zambusi et al, 2022). Interestingly, granulin-dependent clearance of lipid droplets and TDP-43 may be an important difference in the inflammation status of microglia between the species (Zambusi et al, 2022). Our results point to a perhaps even earlier role of microglia in mechanically restoring tissue integrity and thus promoting healing of the brain tissue. Drugs targeting the cytoskeleton could be used to strengthen the potential mechanical role of microglia in tissue contraction (Gao & Nakamura, 2022). This might be similar to approaches in which axon regeneration after spinal injury is enhanced by drug-stabilising the actin cytoskeleton (Hellal et al, 2011; Ruschel et al, 2015).

The accumulation of microglia has also been observed in other CNS injury contexts, for example, in stroke (Lalancette-Hebert et al, 2007) and spinal cord injury in mice (Zhou et al, 2020, Brennan et al, 2022), suggesting wider relevance of our findings. Similar to our observations, microglial and/or macrophage accumulations are progressively surrounded by a dense meshwork of astrocytic processes in these models, in a process termed corralling (Wanner et al, 2013). Loss of microglia or microglia/macrophage-specific gene knockout of adhesion molecules, such as plexin-b2, leads to disorganised and increased wound size (Lalancette-Hebert et al, 2007, Zhou et al, 2020). Conversely, inactivating the signalling molecule Stat3 in astrocytes leads to similar phenotypes, suggesting interactions between astrocytes and microglia (Wanner et al, 2013). The exact role of microglia in wound compaction or corralling is unclear in these injury models. Here, we have established an injury system in zebrafish where corralling-like cellular behaviour can be directly observed and manipulated.

In conclusion, we provide in vivo evidence that forces generated by microglia could be crucial to contracting injured brain tissue and thus facilitating subsequent repair processes. Ultimately, our results may inspire therapeutic approaches targeting mechanical forces related to microglia and other cell types for healing traumatic brain injuries and other tissue damage.

# Materials and Methods

## Fish husbandry

All zebrafish lines were kept and raised under standard conditions (Westerfield, 2000) and all experiments were approved by the UK Home Office (project licence no.: PP8160052) or according to German animal welfare regulations with the permission of the Free State of Saxony (project licence no.: TVV36/2021). Following the guidelines of the 3Rs, we only used larvae aged up to 5 dpf. For experimental analyses, we used larvae of either sex of the following available zebrafish lines: Tg(Xla.Tubb:DsRed)[zf148] (Peri & Nüsslein-Volhard, 2008); Tg(betaactin:utrophin-mCherry)[e119] (Compagnon et al, 2014); Tg(h2a.F/Z:GFP)[kca6] (Pauls et al, 2001) (referred to as Tg(h2a:GFP)); Tg(mpeg1.1:GFP)[gl22] (Ellett et al, 2011); Tg(mpeg1.1:mCherry)[gl23] (Ellett et al, 2011); Tg(irf8)[st95] (Shiau et al, 2015); Tg(elavl3:MA-mKate2)[mps1] (Tsata et al, 2021). The Tg(her4.3:GFP-F)[mps9] transgenic zebrafish line has been previously described by Kolb et al (2023) and was established using the DNA constructs and methodology described below. If necessary, larvae were treated with 100 M Nphenylthiourea (PTU) to inhibit melanogenesis. All chemicals were supplied by Sigma-Aldrich unless otherwise stated.

## Generation of Tg(her4.3:GFP-F) transgenic fish

To create the donor plasmid for generation of her4.3:GFP-F transgenic zebrafish, the sequence coding for the membrane-localised GFP (EGFP fused to farnesylation signal from c-HA-Ras) was amplified from the pEGFP-F vector (Clonetech) using oligos 5′-TTATTTATCGATCCACCATGGTGAGCAAGGGC-3′ and 5′ TTTATTATC-GATTCAGGAGAGCACACACTTGCAGCT-3′ and cloned downstream of the her4.3 (previously known as her4.1) zebrafish promoter (Yeo et al, 2007). Transgenic fish were established by injection of 40 pg of the donor plasmid together with mRNA of the Tol2 transposase into one-cell embryos (Suster et al, 2009).

## gRNA injections

The gRNAs were injected into the yolk at the one-cell stage of development. The injection mix was prepared on the morning of injections. The mix consisted of 1 liter Cas9 protein (M0369M; BioLabs), 1 liter Fast Green FCF dye (235345-9; Sigma-Aldrich), 1 liter 250 ng/liter SygRNA-tracr (TRACRRNA05N; Sigma-Aldrich), 1 liter gRNA, and 1 liter nuclease-free water. When two gRNAs were co-injected, the nuclease-free water was substituted with the second gRNA. After mixing gRNAs and tracr (and water if using), the mixture was heated to 95 degrees for 5 min and then kept on ice for 20 min. After this, the Cas9 and dye are added, and the mixture is again heated to 27 degrees for 10 min. For every experiment, two injection mixtures were

n = 6), blebbistatin-treated microglia contacting astrocytes (blue, n = 3), microglial migration (red, n = 8), and astrocytes that were contacting microglia (yellow, n = 10). Area change represents the difference between T(n + 1) and Tn as shown in (B) as magenta and blue, respectively. **(E)** Cumulative area change for astrocytes with homotypic (green, n = 8) or microglial (red, n = 6) contacts after ablation. Area change represents the total area changed between T0 and the given time point. Statistics were assessed by two-way ANOVA with Šídák's multiple comparison test, *P < 0.05, **P < 0.01. All scale bars represent 10 μm, and all ablation sites are marked with yellow boxes on montages.

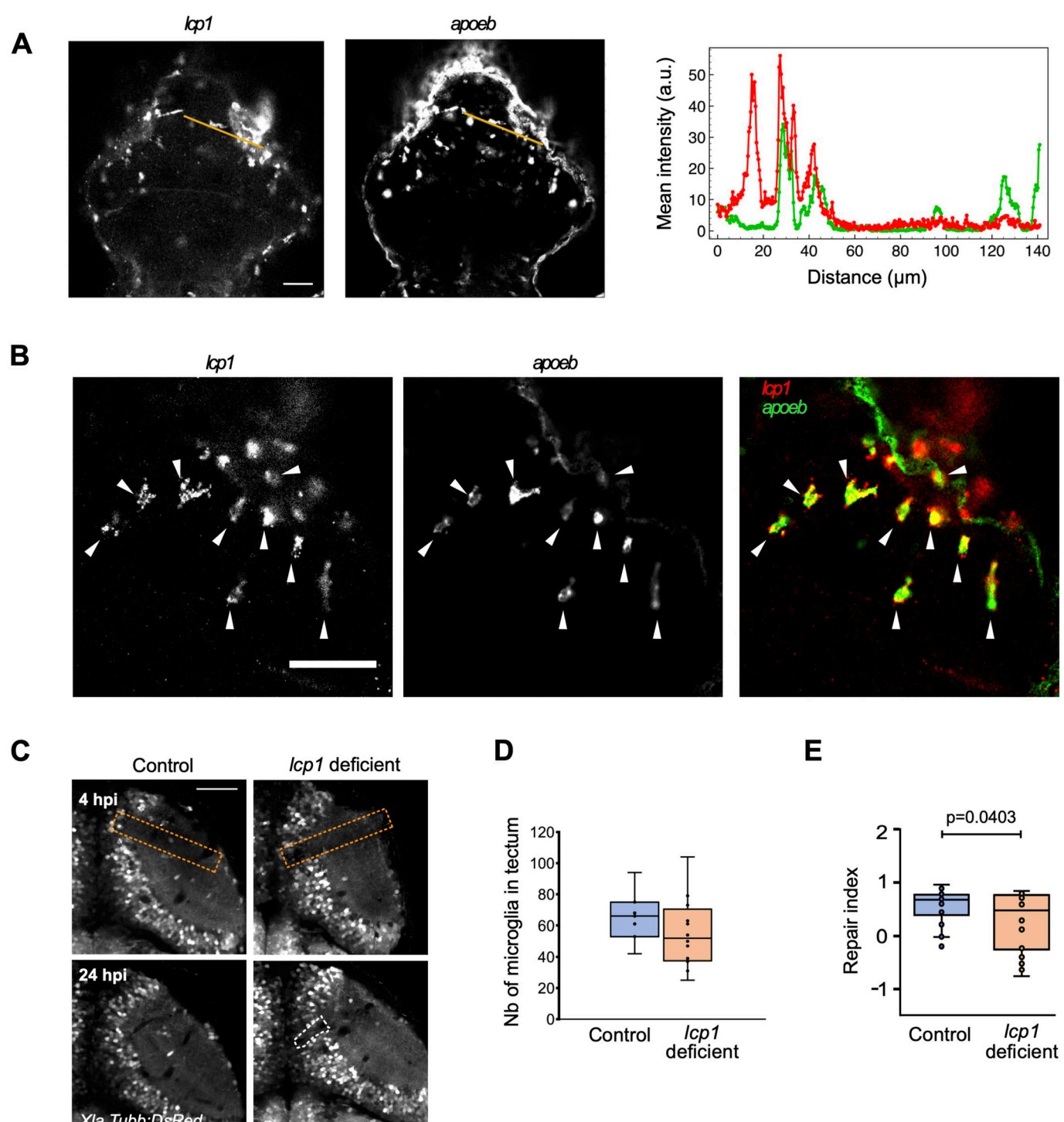

**Figure 7. Microglial action relies on *lcp1*.**
**(A)** Expression of lcp1 and apoeb as detected by hybridisation chain reaction fluorescence in situ hybridisation in injured larvae at 6 hpi. The intensity profiles shown on the graph (red: lcp1; green: apoeb) were measured along the orange lines shown in the images. **(B)** Zoomed-in hybridisation chain reaction fluorescence in situ hybridisation images for lcp1a (green) and apoeb (red), showing an almost complete overlap of patterns. Arrows point to lcp1+/apoeb+ cells at the lesion site. **(C)** Representative images of wound closure in control and haCR-injected animals are shown; orange dashed lines: injury site; white dashed lines: wound delimitation in the PVZ. **(D)** Boxplot representation of the repair index in all haCR-injected animals (*P* from an unpaired *t* test). **(E)** Boxplot representation of the number of microglia in the optic tectum at 24 hpi in haCR-injected animals. All *t* tests were non-significant (*P* > 0.05). For all experiments, n = 4 for control-injected, and n = 10 for haCR-injected animals. All scale bars represent 50 *µ*m.

**Table 1.  Primer sequences for lcp1.**

| Primer | Sequence | Enzyme |
|--------|----------|--------|
| Forward | tgaccttgtcctgcagATGT | BslI |
| Reverse | AAGGTGATCTTCCCGTCCTG | BslI |

made, one with the gRNA of interest and one with a control gRNA (5′-TTACCTCAGTTACAATTTAT-3′). lcp1 was targeted with a gRNA (5′-GAACCCGGUACCCCGGCAGA-3′) as previously published (Keatinge et al, 2021).

### RFLP analysis of gRNA efficiency

To test the efficacy of gRNAs targeting lcp1, we used the RFLP method as described in Keatinge et al (2021). See Table 1 for primer sequences and corresponding restriction enzymes.

Genomic DNA was extracted from individual embryos (1–5 dpf) by heating the whole embryo with 100 liter of 50 mM NaOH at 95°C for 10 min followed by the addition of 10 $\mu$l of 1 M HCl. PCR products were generated using BioMix Red (Bio-25006; Bioline) and were subsequently incubated with the respective restriction enzyme (BioLabs) for 1.5 h at the specified temperature, followed by separation on a 2% agarose gel (100 V for 40 min) and visualisation on a transilluminator.

### Hybridisation chain reaction fluorescence in situ hybridisation (HCR FISH)

For HCR FISH, HCR probe sets were ordered from Molecular Instruments, targeting the following mRNAs: apoeb1 (NCBI Reference NM_131098.2) and lcp1 (NCBI Reference NM_131320.3). All HCR buffers and the h1 and h2 hairpins were purchased from Molecular Instruments. HCR FISH was performed as previously described in Choi et al (2018). Larvae were fixed at 6 hpi in 4% PFA at RT for 3 h. Fixed larvae were dehydrated in increasing concentrations of methanol and stored at −20°C in methanol overnight. Larvae were rehydrated in decreasing concentrations of methanol and washed 3 times in 1x PBST. Permeabilisation was performed by incubating the larvae for 40 min in 30 $\mu$g/ml Proteinase K (in 1x PBST). Larvae were post-fixed in 4% PFA at RT for 15 min and subsequently washed 4 times in 1x PBST. Pre-hybridisation was performed with pre-warmed HCR hybridisation buffer (Molecular Instruments) for 30 min at 37°C. 2 pmol of the respective HCR probes was diluted in 500 $\mu$l HCR hybridisation buffer. Hybridisation buffer was replaced with the HCR probe mix, and hybridisation was performed for 16 h at 37°C. To remove excess probe, the larvae were first washed four times for 15 min each in HCR wash buffer (Molecular Instruments) at 37°C, followed by several washes in 5X saline–sodium citrate buffer with 0.1% Tween-20 (SSCT) at RT. Subsequently, larvae were incubated in HCR amplification buffer (Molecular Instruments) for 30 min at RT. Meanwhile, 30 pmol of the respective hairpin h1 and h2 (Molecular Instruments) was prepared by snap-cooling 3 $\mu$l of the respective 10 $\mu$M hairpin stock. Snap-cooling was performed by incubating the hairpins for 90 s at 95°C and cooling them down for 30 min at RT in the dark. Hairpin solution was prepared by transferring 10 $\mu$l of the snap-cooled h1 and h2 hairpins to 500 $\mu$l amplification buffer. The amplification step was performed by incubating the larvae for 16 h in hairpin solution. Excess hairpin solution was removed by washing the larvae four times in 5x SSCT for 15 min each at RT. Larvae were stored in 70% glycerol (in 1x PBST) at 4°C.

### Induction of brain injury

For inducing brain injuries, Zebrafish larvae at 4 dpf were anaesthetised using 0.016% ethyl 3-aminobenzoate methanesulfonate (MS-222; Sigma-Aldrich). They were then mounted in 1% low melting point agarose (LMPA, Life Technologies) with their dorsal side facing upward and their anteroposterior axis as horizontal as possible. The optic tectum was then injured under visual guidance via a stereomicroscope using a metal insect pin with a diameter of 80 $\mu$m (Fine Science Tools) mounted on a micromanipulator (Narishige or WPI). The tip of the metal pin was slightly bevelled to facilitate penetration of the skin, and the angle between the pin and the horizontal plane was 20–30°. For induction of injury, the tip of the metal pin was inserted into the optic tectum to a depth of 200 $\mu$m and immediately retracted. The injury procedure itself took less than 2 s. After an injury, larvae were carefully released from the agarose and allowed to recover in fresh fish water for varying amounts of time depending on experimental requirements (typically 4 h before imaging).

### Blebbistatin treatments

Animals were incubated with 1 $\mu$M para-nitroblebbistatin (Axol Bioscience or BIOZOL), referred to as blebbistatin, solubilised in double-distilled $H_2O$. The drug was added directly to the fish water 7 h after the injury, and incubation continued until the time of experimental readout. The repair index was determined as described in the dedicated section.

### KI20227 treatments

KI20227 (Tocris) powder was dissolved in sterile-filtered DMSO to produce a 50 mM stock. 2 dpf Tg(Xla.Tubb:DsRed); Tg(mpeg1:GFP) larvae were screened for transgenes, anaesthetised, and transferred into wells with 500 $\mu$l of a 25 $\mu$M solution of KI20227 diluted in fish water with PTU. Larvae were incubated for 48 h before injury or cell counting to verify the efficient suppression of microglia in the optic tectum (Fig 4H).

### Immunofluorescence

All incubations were performed at room temperature unless stated otherwise. At the time point of interest, larvae were fixed in 4% PFA-PBS containing 1% DMSO at 4°C overnight. Larvae were then washed in PBTx (1% Triton X-100 in PBS). After permeabilisation by incubation in PBTx containing 10 $\mu$g/ml Proteinase K for 15 min, larvae were washed twice in PBTx. Then, larvae were incubated for 1 h in 4% BSA-PBTx (blocking buffer) and incubated with primary antibody (mouse anti-HuC or donkey anti-Myh10) diluted at 1:100 in blocking buffer at 4°C overnight. On the next day, larvae were washed six times in PBTx for 20 min each, followed by incubation with secondary antibody (anti-mouse Cy3, Jackson) diluted at 1:300 in

blocking buffer at 4°C overnight. The next day, larvae were washed six times in PBTx for 30 min each and twice in PBS for 15 min each, before mounting in 1% low melting point agarose for imaging.

## HuC/EdU staining

Brain injuries were carried out on 4 dpf Tg(h2a:GFP) larvae as described above. After performing the lesions, larvae were immediately placed in a solution of 50 $\mu$m 5-ethynyl-2'-deoxyuridine (EdU; Sigma-Aldrich) and were allowed to develop in this solution.

At 5 dpf (24 hpi), the larvae were culled with an overdose of MS-222 and fixed with 4% PFA overnight on a rocker. Larvae were washed three times with PBTx and incubated in methanol for 5 min before being placed in fresh methanol at −20°C overnight. Samples were rehydrated in a dilution series (75, 50, 25%) from methanol to PBTx and then washed in PBTx. Larvae were digested using Proteinase K (10 $\mu$g/ml in PBTx) for 45 min at room temperature. They were then fixated for 15 min in 4% PFA at room temperature. After three washes in PBTx, the animals were incubated in a 1% DMSO/0.5% Triton X-100 solution for 20 min and washed in PBTx. The EdU Click-iT reaction solution (Roche) was prepared according to the manufacturer's instructions, and larvae were incubated at room temperature in the dark for 2 h. All further incubations, including the immunofluorescence, were carried out in the dark. Larvae were thoroughly washed several times in PBTx before proceeding to immunofluorescence with anti-HuC to identify neurons.

## Image acquisition

Before live confocal imaging, zebrafish larvae were anaesthetised using 0.008% MS-222 and mounted in 1% LMPA, with their dorsal side facing upward to image the optic tectum. The agarose was then covered in fish water with 0.008% MS-222 to prevent desiccation during imaging and animal movements.

For long-term imaging, the larvae were mounted in 1% LMPA in a 90-mm Petri dish and placed on the microscope stage using a custom 3D printing adapter. The animals were maintained at a stable temperature of 28°C using a stage incubator with a bucket of water to prevent excessive evaporation of the fish water from the Petri dish. For z-stacks of the optic tectum in living or fixed samples, images were acquired with typically 2-$\mu$m intervals between optical planes. In injured animals, z-stacks were chosen such as to encompass the entire injury site.

For acquiring short time-lapses on living larvae or 3D stack on fixed animals, larvae were mounted in LMPA on a microscope coverslip. A reservoir was obtained by setting the coverslip on a microscope slide with a spacer made of vacuum grease and filled with fish water to keep animals hydrated.

All fluorescence images except for HCR FISH and laser cutting samples were acquired on a Zeiss LSM 880 confocal microscope equipped with an Airyscan detector and a 20x NA 1.0 LWD water dipping objective lens.

Fluorescently HCR FISH–stained larvae were recorded as confocal stacks. Confocal imaging was performed using a Carl Zeiss LSM 980 inverse laser scanning microscope. A 20x Plan-Apochromat, Air, DIC objective (Zeiss, NA = 0.8) was used. The 561-nm and the 633-nm laser lines were used for confocal microscopy. The pinhole size was adjusted to 1 Airy Unit.

## Repair efficiency assay

The larvae were injured and imaged as described. After imaging at 24 hpi, the larvae were culled and genotyped individually by analysis of the RFLP to identify WT or irf8$^{-/-}$ animals.

For quantification of injury volume, the region of the PVZ that displayed signs of damage was manually outlined in ImageJ (https://imagej.net) plane after plane. The volume was reconstituted using a custom script (Supplemental Files) and quantified using the plugin 3D geometrical measure (https://imagejdocu.list.lu/plugin/analysis/3d_analysis/start/) (Ollion et al, 2013).

The repair index (RI) was estimated by calculating the following ratio: RI = 1-V24 hpi/V4hpi.

## Trajectory analysis

Mean-squared displacement calculations were made with DiPer (Gorelik & Gautreau, 2014) using data obtained from MTrackJ. Trajectory characteristics such as straightness or z-displacement were manually extracted from MTrackJ data.

The discrete Fréchet distance used to compare microglial and neuron trajectories in Fig 3J was calculated with a custom script based on the library "trajectory distance" (https://github.com/bguillouet/traj-dist).

The trajectory anisotropy graph is shown in Fig 1I and was created using a custom script (Fig S2, Supplemental Files). Displacement vector intersections from Fig 3 were determined using a custom script (cf Supplemental Files).

## Image processing and analysis

The wound closure kinetic was estimated by measuring the fluorescence intensity in an ROI delimiting the initial wound area in the PVZ over time. Long-term imaging data were registered for 3D motion in Fiji using the plugin 3D drift (Parslow et al, 2014).

Before tracking, time-lapse images of Tg(h2a:GFP) were processed to improve the identification of each nucleus in Fiji using LocalNormalization (Sage & Unser, 2001) and denoising using the PureDenoise (Luisier et al, 2010) plugins. Manual tracking of cells was done using the Fiji plugin MTrackJ (Meijering et al, 2012).

The quantification of microglial accumulation kinetic was done by measuring the mean fluorescence intensity in the neuropil over time using ImageJ. The rostrocaudal extend curve was estimated by manually drawing a polyline starting rostrally and passing at an equal distance between the PVZ external and internal boundaries. Microglial accumulation positions were estimated from manually defined cell centroids in ImageJ after normalisation of the tectum to consider variations in size and position depending on the animal (Fig S3). Before estimation of the repair index in Tg(Xla.Tubb:DsRed); Tg(irf8+/−) fish, images were blinded using a custom script (Supplemental Files). Image deconvolution for Fig 5 was performed using DeconvolutionLab2 (Sage et al, 2017) in Fiji and a 2D calculated point spread function using the Born–Wolf algorithm thanks to the point spread function generator Fiji plugin (Sage et al, 2017). Microglial morphology and endosome analysis were performed by

manually drawing cell contours and measuring endosome diameters using Fiji.

### Laser severance and quantification

Cell process ablation in tectum-injured larval zebrafish Tg(her4.3:GFP-F); Tg(mpeg1:mCherry) was completed using a confocal microscope (SP8 inverse, Leica). Zebrafish larvae were mounted in 0.5% LMPA as previously described in a multiwell cell culture imaging chamber (Ibidi) with the top of their heads touching the coverslip surface at 1 dpi. The imaging was completed in a temperature-controlled incubation chamber (25°C), using a 25x/0.95 HC FLUOTAR L water immersion objective (Leica). Single-photon imaging was completed using a white light laser tuned to 488 nm (10% laser power) and 594 nm (2% laser power), with the emission range of the built-in HyD detectors set to the emission range of GFP and mCherry (respectively). Ablation was completed using a 5 × 1 $\mu$m ROI with a 2-photon laser tuned to 800 nm. All images were acquired at 14x zoom, with 10 planes (1-$\mu$m spacing). The ROI was exposed to the laser for all 10-plane images (512 × 512 pixels), with the focal point of the ROI placed on the astrocytic contact with the extended microglia during the full-stack acquisition (15 s). This was followed by a time series recording (5 min, 15-s interval between frames, 512 × 512 pixels) to assess the movement of astrocytes and/or microglia. High-speed recordings were acquired for one optical section using the same imaging parameters, with a 0.65-ms interval between frames. Contacts between astrocyte and microglia were selected based on the extension of astrocytic processes and microglial membranes. Laser-induced migrating microglia were those which were less than 10 $\mu$m away (2x the ablation ROI width) from the injury site at the time of laser activation. Astrocyte–astrocyte contacts were selected within the optic tectum adjacent to the injury site, but not touching it (circa 20 $\mu$m into the tectum from the internal edge of the injury site). For 3D images, the microglia and astrocytic processes were segmented manually in Fiji, using maximum intensity projections for 3D time series. The area change between frames was calculated from these ROI sets for either the microglia or astrocytes. For 2D images, the microglia were manually contoured using the built-in segmentation editor in FIJI (Schindelin et al, 2012). The centre of the ablation ROI was calculated by applying a 5 × 1 $\mu$m ROI over the ablation site after rotating the image until the ablation site was horizontal. The relative edge distance of the microglial ROI to the centre point of the ablation site was quantified by comparing the centre of the ablation site with the leading edge of the microglia.

### Grey value distribution measurements for HCR signal

To measure the grey value distribution of the HCR signal within the lesioned tectum (Fig 7), a single representative optical section (1 $\mu$m) was selected from a recorded confocal z-stack, encompassing expression within the lesion site. Pictures were converted into 8-bit greyscale, and levels were adjusted by histogram clipping. Grey value distribution was measured along a 140-$\mu$m line (width 6 $\mu$m) originating at the tectal lesion site and extending within the unlesioned tectum. HCR signal grey values were measured using the Plot Profile tool in Fiji/ImageJ.

### Numerical simulations and programming frameworks

Simulations were performed using the PhysiCell framework (Ghaffarizadeh et al, 2018) on a MacBook Pro (2.6 GHz 6-core i7 CPU, 16 Gb RAM) after compiling the custom C++ source code using GCC 11.0.3 (cf Supplemental Files).

All Python custom scripts were written using Spyder 4.1.2 and ran with Python 3.7.6 using the Anaconda3 environment.

### Statistical analysis

Curve fitting was done using the Curve Fitting Toolbox in MATLAB. The t tests were performed using Excel (Microsoft). Two- and three-factor nested ANOVA tests were performed according to Reference (50) using Excel. Before estimation of the repair index in Tg(Xla.Tubb:DsRed); Tg(irf8+/−) fish, images were blinded using a custom script (Supplemental Files).

### Viscoelastic model of neuron dynamics

We considered a model where neurons were attached to springs in a viscous medium (cf Fig 2F). The position of a neuron can be obtained by solving Newton's first law of motion for a harmonic oscillator with an additional viscous force:

$$m\frac{d^2x}{dt^2} + v\frac{dx}{dt} + kx = 0$$

which gives

$$x(t) = ae^{-vt/2}\cos\left(\sqrt{\frac{k}{m} - \frac{v^2}{4}}\, t - \varphi\right)$$

We used this equation to fit the experimental fluorescence intensity curves shown in Fig 1D using MATLAB.

### Optic tectum multi-agent theoretical model

The model we developed is based on the work from Ghaffarizadeh et al (2018). We adapted the force balance and did not consider the interactions with the microenvironment. We also neglected cell cycle dynamics as it did not appear to play an essential role from our experimental observations (limited cell proliferation/neurogenesis). The agents can exert/sense 4 forces:

(1) Cell–cell adhesion
(2) Cell–cell repulsion
(3) Cell–cell elastic traction
(4) Medium/ECM drag force

These forces are used in the computation in the form of gradients to update the velocity of each agent depending on its nature and position relative to every other agent. The skin agent has a particular role as it is not active or mobile. It was added to model the stiffness of the skin, which affects cell and tissue dynamics and the overall force balance. The shape of the PVZ was obtained from microscope image processing. Neurons were modelled as low mobility cells as observed

**Table 2. Summary of the parameters used for the different agents in the simulations for the present work.**

| Parameter | Neurons | Microglia | Skin cells | Biophysical meaning |
|---|---|---|---|---|
| Persistence time (min) | 10 | 10 | 0 | Directionality persistence of mobile cells |
| Migration speed (micron/min) | 0.01 | 1 | 0 | Cell velocity (related to Floc) |
| Relative repulsion | 5 | 5 | 5 | Cell–cell repulsion force amplitude |
| Relative adhesion | 0.1 | 0 | 0 | Cell–cell adhesion force amplitude |
| Elastic coefficient(1/min) | 5.10−7 | 5.10−7 | 5.10−7 | Amplitude of the elastic forces (Ftrac) |

experimentally on intact animals: a slow random motion without migration bias. Microglia were considered more motile, with an extra elastic force exerted between them and with neurons.

The elastic traction force is a phenomenological representation of forces exerted by microglia on astrocytic processes and the extracellular matrix. This allows for a simplified model able to reproduce the main observations with only a few parameters.

Neurons are represented only by their soma. We considered that the mechanical component of their protrusions is implicitly considered by the elastic traction force.

Below is the formulation of the agent velocity:

$$\vec{V_i} = \frac{1}{\eta} \sum_{j \in N(i)} \left[ \vec{F}^{ij}_{cca} + \vec{F}^{ij}_{ccr} + \vec{F}^{ij}_{trac} + \vec{F}^{i}_{loc} \right]$$

where Fcca is the cell–cell adhesion force; Fccr, the cell–cell repulsion force; Floc, the locomotion force (for motile cells); and Ftrac, the elastic traction force. $\eta$ is the medium effective viscosity.

The traction force is exerted only between microglia and other cells or between themselves. It follows Hooke's law and can be written as follows:

$$\vec{F}^{ij}_{trac} = \begin{cases} -k \left( \vec{r_j} - \vec{r_i} \right), & \text{if } i, j \text{ are a microglia} \\ \vec{0}, & \text{if } i, j \text{ are other types of cell} \end{cases}$$

### Numerical implementation of the model

We used the PhysiCell (version 1.7.1) framework for all simulations (http://PhysiCell.MathCancer.org). The program flow is detailed in Ghaffarizadeh et al (2018).

Briefly, the velocity is updated for each agent, depending on its nature, its position, the forces exerted, and the position of other agents interacting with it.

Table 2 summarises the parameters used for the different agents in the simulations for the present work.

The distribution of the different cell types was determined from fluorescence microscopy images using manual delimitation of the tectum, PVZ, and skin outline.

The positions of cells were generated using a Python script computing close-packing disc distribution inside an arbitrary 2D geometry.

For neurons, the actual shape of the PVZ was used. For skin cells, the thickness was artificially increased to mimic the skin

stiffness. For microglia, positions were set manually by drawing ROI on the image mask in the neuropil region.

The generated files were used as inputs for the simulation program, which generated a series of images at different time points. We used those images to generate the simulated wound closure kinetics in Fig 4C.

## Data Availability

The data that support the findings of this study are available from the corresponding author upon reasonable request.

## Supplementary Information

## Acknowledgements

We thank David Greenald (CRH, University of Edinburgh) and Katy Reid (CDBS, University of Edinburgh) for the kind gift of transgenic fish. We thank Jason Early (CDBS, University of Edinburgh) for his help with fluorescence microscopy. We thank Themistoklis M Tsarouchas (SCRM, University of Edinburgh) for his help with genotyping. We thank Paul McKlin and Randy Heiland (Indiana University) for their help with the use of PhysiCell. We thank the uCreate Maker Space from the University of Edinburgh for 3D printing. This work was supported by the CMCB Light Microscopy Facility, a core facility of the CMCB Technology Platform at the Technische Universität Dresden. The laser ablation experiments and HCR imaging were performed on Leica SP8 and the Andor Dragonfly (respectively) of the CMCB Light Microscopy Facility, a core facility of the CMCB Technology Platform at the Technische Universität Dresden. This work has been supported by the Biotechnology and Biological Sciences Research Council (BB/S0001778/1) (to T Becker, CG Becker, and F El-Daher), Moray Endowment Fund (to F El-Daher), and Alexander-von-Humboldt Professorship Award (to CG Becker).

### Author Contributions

F El-Daher: conceptualisation, data curation, software, formal analysis, supervision, funding acquisition, validation, investigation, visualisation, methodology, project administration, and writing—original draft, review, and editing.
SJ Enos: investigation, visualisation, methodology, and writing—review and editing.
LK Drake: investigation and methodology.

D Wehner: resources.

M Westphal: investigation.

NJ Porter: investigation.

CG Becker: conceptualisation, supervision, funding acquisition, methodology, and writing—review and editing.

T Becker: conceptualisation, supervision, funding acquisition, methodology, and writing—review and editing.

## Conflict of Interest Statement

The authors declare that they have no conflict of interest.

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
