## [Reviewer comments · Life Science Alliance]

Life Science Alliance

Microglia are essential for tissue contraction in wound closure after brain injury in zebrafish

Francois El-Daher, Stephen Enos, Louisa Drake, Daniel Wehner, Markus Westphal, Nicola Porter, Catherina Becker, and Thomas Becker

DOI: <https://doi.org/10.26508/lsa.202403052>

Corresponding author(s): *Francois El-Daher, University of Edinburgh*

Review Timeline:	Submission Date:	2024-09-20
	Editorial Decision:	2024-09-23
	Revision Received:	2024-10-07
	Accepted:	2024-10-08

Transaction Report:

Please note that the manuscript was previously reviewed at another journal and the reports were taken into account in the decision-making process at Life Science Alliance.

Point-by-point rebuttal letter

for the manuscript “Wound closure after brain injury relies on force generation by microglia in zebrafish” by El-Daher F. et al.

Reviewer #2 (Remarks to the Author):

To prove that microglia indeed exert mechanical force to resolve brain injury, the author performed additional laser-cutting experiments, in which they found that after laser severing the contacts between microglia and astrocytes, microglia immediately retract their processes. Although the author argued that the retraction of microglia processes is due to the relax of mechanical tension between microglia and astrocytic processes, it might be simply caused by the response of microglia to avoid being harmed by laser.

RESPONSE: We observe a dynamic of retraction ~6 times faster (0.89 $\mu\text{m}/\text{sec}$) than previously published values for microglia process retraction *in vivo* in mice (0.144 $\mu\text{m}/\text{sec}$). Hence, the extremely fast retraction or recoil of the microglia is difficult to explain by an active avoidance movement of the cell, which is limited by the timing of cytoskeletal dynamics. We discuss this point on p. 8. This is consistent with the presence of mechanical tension between microglia and astrocytic processes.

COMMENT: Second, enrichment of Myh10 and beta-actin:utrophin in microglia and the defective wound healing after lcp1 disruption is not sufficient for authors to argue for the involvement of these cytoskeleton gene in mechanical force generation, as cytoskeleton rearrangement is also required for microglia migration and efferocytosis.

RESPONSE: We take enrichment of labelling as correlative evidence that microglia have the potential to generate forces. The fact that other cellular movements require these proteins does not contradict this modest conclusion. We scrutinized the manuscript to remove any potential over-interpretation.

COMMENT: Finally, the wound healing defect caused by sgRNA/Cas9 transient knockout/knockdown of lcp1 is quite marginal.

RESPONSE: The effect of LCP1 disruption is highly robust, as also indicated by our re-analysis of the data asked for by reviewer #4. We removed from the analysis animals with fewer microglia in lcp1-deficient animals (<50 cells): the repair index distribution is still statistically different compared to WT animals ($p=0.018$). Figure 7 has been updated to show the improved analysis.

The manipulation is highly selective microglial, the only immune cell type present in the injury site. Due to the specific role of lcp1 in cytoskeletal force generation we weaken the microglial cells by disrupting the gene. We would not expect this strategy to have the same effect size as removing microglia altogether.

COMMENT: This is potentially problematic when we consider the off-target effect of sgRNA/Cas9 and the frequently-observed inconsistency of phenotypes between Crisprant and stable mutant.

RESPONSE: Creating somatic mutants with highly active CrRNAs is a relatively new method established independently in two laboratories (PMID: 33416493, PMID: 33914736) and is now widely used in zebrafish, e.g. PMID: 39147780. We use internal controls in every experiment to show that a targeted restriction site is completely disrupted by the highly active CrRNA (Fig. S8). Creating a stable germline mutant would take at least half a year and comes with its own problems, e.g. background mutations. So far, we have not observed differences between somatic and germline mutants in our work and are not aware of any such discrepancies in the literature.

Differences between acute manipulations and germline mutants existed in the past for an unrelated method in zebrafish that is now outdated. Anti-sense morpholinos were used to transiently block translation of genes and differences in phenotypes were indeed frequent (PMID: 29049395). The method we used is robust and used at scale without phenotypic differences (PMID: 37968389).

COMMENT: Due to above reasons, we do not think the appealed version of manuscript addressed the concerns from us, therefore it is not suitable for publication in Nature Communications.

RESPONSE: We regret that we were unable to convince this reviewer, but think that our new changes further strengthened our manuscript providing evidence for a potentially new role for microglia in brain wound healing.

COMMENT: The major concern raised by reviewer 3 is “Mechanistically, the study continues to confuse correlation with causation, leading to overstatements and biologically unfounded assertions”, which is similar with ours, i.e., “ the causal relationship between mechanical forces generated by microglia and wound closure”.

RESPONSE: We scrutinized the manuscript for potential over-statements and have moderated our language in the new manuscript.

COMMENT: As mentioned before, we do not think the laser-cutting and transient lcp1 knockdown/knockout experiments provide sufficient evidence to address this concern.

RESPONSE: For the above reasons, we find that high speed of process retraction after laser cutting and high specificity and robustness of lcp1 gene disruption, together with our other genetic and modelling approaches, are difficult to interpret without considering force generation by microglial processes. This is a previously undescribed potential role for these cells in CNS wound healing.

Reviewer #4 (Remarks to the Author):

Summary:

The authors discovered a novel mechanism of how microglia and F-actin system quickly close the wound on optic tectum in zebrafish embryo. It offers many interesting experiments and provides unique insights. However, I must raise the following concerns.

Major:

In most part of the paper, the evidence is well presented, and most of the conclusions are well drawn, but there are lines that are arguably leaping in logic.

In line 277 and line 298 “, the authors should use “potential pulling force” and “potential contraction force” since the paper did not, at least not at this point, establish what kind of mechanical force is driving the movement.

RESPONSE: We thank the reviewer for agreeing that our “conclusions are well drawn” and we changed the phrasing as suggested.

In line 320, “Overall, these results support the hypothesis that...”, is an overstatement. The observation is at best “aligned” or “not conflicting” the hypothesis of any elastic force involving. Again, the paper did not establish what kind of force is driving the movement.

RESPONSE: We thank the reviewer for these suggestions. Our data is indeed best explained by a role of microglia in generating mechanical force. All our experiments are performed in vivo, which is a strength, but this approach does not lend itself easily to more direct measurements of force. For these reasons, we adopted the more moderate language suggested by this reviewer.

In line 390, the experiment do not “confirm” the tension-force driven simulation. It does established the important role microglia plays, but not the tension-force.

RESPONSE: We agree with the reviewer and now write “Our results are consistent with...”

In line 440, this particular set of experiment result does not provide useful metric to draw conclusion. Potential metric here can be how fast is the “tethering” movement comparing to its surrounding random movement. Is the contact only being pulled or it is being pushed at other sites at the same time? How far or how fast is the tethering need to pull to be considered as a SAT event in line 432 and 435? (Further, is any of the knock-out experiments changing those metric?)

RESPONSE: We only observed contacts being pulled by microglia during SAT events. The whole process typically lasts 2 min with a displacement of 1 μm . Smaller displacements due to microglia movements weren't considered as SAT events. There is very little other displacement of astrocytic processes as the meshwork appears static at this time-scale (~ 30-40 min). We agree with the reviewer that astrocytic processes could be pushing microglia. However, our laser-cutting experiments show that the microglial processes are under tension. This is inconsistent with a pushing model. We acknowledge that possibility now in the manuscript on p. 11.

COMMENT: In line 497, if I am not wrong, any cell with processes will retract if the process is severed or damaged.

RESPONSE: This is probably true, but would need an active cytoskeletal re-arrangement, implying widespread and instantaneous activation of ADF/cofilin across a broad swath of the microglia cell. These events are several times slower than the recoil reaction we see. For example, we note in the text of the paper (line 478) that the microglia processes in the high-speed recordings recoil away from the cut 5 times faster than migration by the same cells (0.89 $\mu\text{m}/\text{sec}$ versus 0.18 $\mu\text{m}/\text{sec}$). Migration speed is consistent with previously published values for microglia process extension and retraction *in vivo* in mice (0.128 $\mu\text{m}/\text{sec}$ and 0.144 $\mu\text{m}/\text{sec}$, respectively) [Bernier 2019]. Hence, the fast retractions we observe after laser-cutting are consistent with the presence of tension on the process at the time of cutting. We make this clearer now in the discussion.

Ref: Bernier et al., 2019, Cell Reports. Figure 2:

<https://www.cell.com/action/showPdf?pii=S2211-1247%2819%2930621-7>

COMMENT: Even if it can be an evident of microglia is stretched by engaging with astrocyte processes, it does not lead to the conclusion that this stretching force leads to the later deformation.

RESPONSE: we agree with the reviewer that our results here are of a correlative nature. Deformation of the astrocytic network and recoil of microglia are highly correlated in space. Moreover, if microglia force generation is weakened by *Icp1* gene disruption wound closure is impaired. This experiment together give an indication that force generation by microglia is on a scale that affects brain wound closure. We make this clear on p. 11.

- Hsu et al, <https://www.ncbi.nlm.nih.gov/pmc/articles/PMC8931656/>
- Le Guyader et al, <https://ashpublications.org/blood/article/111/1/132/107908/Origins-and-unconventional-behavior-of-neutrophils>

- Garcia-Lopez et al, <https://www.nature.com/articles/s41467-023-40662-7>
- Bader et al, <https://www.ncbi.nlm.nih.gov/pmc/articles/PMC8453019/>

In line 559, figure 7D does not show statistical significance (what is the p value?), but it still shows a shift in the average. If you remove the results that have low microglia, can figure 7E still show significant? This experiment is crucial for the hypothesis since this is the only experiment that can tie to mechanical force exerted by microglia (but yet, not necessarily tension, might just be the microglia local mobility that led to other mechanism). The analysis need to be more clean and careful.

RESPONSE: We thank the reviewer for this interesting suggestion. We thus performed the suggested manipulation by analysing only animals with at least 50 microglia present in the injured hemisphere 24h post-injury. Our results show that the difference is statistically significant with a $p=0.018$. The difference in Figure 7D is indeed not significant with a p-value of 0.38. We have updated Figure 7D and E and the text l. 558-561 accordingly.

COMMENT: In short, I think the authors leap to the “tension force” and overshadowed an otherwise air-tight discovery.

RESPONSE: We are happy that the reviewer sees the merit in our study and we hope that with our extra explanations and moderated conclusions our study can give important impulses for future work on the mechanics of microglia.

COMMETN: Minor:

- Fig 4B, the simulation shows all microglia accumulated near the neuron wound that is inconsistent with the experiment, which microglia mostly stay near the pial side. (May indicate the tension hypothesis being flawed)

RESPONSE: The exact localisation of the accumulated microglia indeed depends on the parameters of the model. The model was established to determine whether microglia cells could potentially exert a net force on tissue. We did not aim to reproduce quantitatively the experiment for this factor. This does not change the validity of the model, only the fine-tuning of the parameters that could generate images more similar to the actual images. We mention this limitation now in the manuscript on p. 11 .

- COMMENT: Fig 2F/G, the model is overly simplified, and the final result is a sigmoid curve, which is very common on many mathematical expressions.

RESPONSE: We agree that the model is very simple. Our aim was to give us a direction of investigation and not to develop a complete representation of the system and events happening after injury. We argue that although simple, it has its value for this purpose. We indicate the limitations of the model on p. 11 of the manuscript.

The fact that we obtain a sigmoidal curve cannot be an argument against the validity of the model since, as the reviewer mentions, it is common.

- COMMENT: The author should at least have a paragraph talking about the limitation of this being an embryonic experiment and might not be replicated in adult brain.

RESPONSE: We thank the reviewer for this suggestion. We now more explicitly discuss the limitations of transferring our findings to other systems in the discussion p. 11.

September 23, 2024

RE: Life Science Alliance Manuscript #LSA-2024-03052-T

Dr. Francois El-Daher
University of Edinburgh
The Chancellor's Building
49 Little France Crescent
Edinburgh EH16 4SB
United Kingdom

Dear Dr. El-Daher,

Thank you for submitting your revised manuscript entitled "Microglia are essential for tissue contraction in wound closure after brain injury in zebrafish". We would be happy to publish your paper in Life Science Alliance pending final revisions necessary to meet our formatting guidelines.

- please be sure that the authorship listing and order is correct
- please upload your main manuscript text as an editable doc file
- please upload your main and supplementary figures as single files
- please upload your table files as editable doc or excel files
- please upload your graphical abstract as a separate file with the file designation, Graphical Abstract
- please consult our manuscript preparation guidelines <https://www.life-science-alliance.org/manuscript-prep> and make sure your manuscript sections are in the correct order
- please add a running title and a summary blurb/alternate abstract for your manuscript to our system
- please add the Twitter handle of your host institute/organization as well as your own or/and one of the authors in our system
- please use the [10 author names, et al.] format in your references (i.e. limit the author names to the first 10)
- please add figure callouts for Figure 4 J,K,L and Figure S4A to your main manuscript text; please edit the Figure callout Figure S8A because this figure only has one panel, we don't need it distinguished with a letter
- please incorporate the Supplemental methods into the main Materials & Methods section

Figure Check:

- please add scale bars to figure S6C

A. FINAL FILES:

-- Summary blurb (enter in submission system): A short text summarizing in a single sentence the study (max. 200 characters including spaces). This text is used in conjunction with the titles of papers, hence should be informative and complementary to

the title. It should describe the context and significance of the findings for a general readership; it should be written in the present tense and refer to the work in the third person. Author names should not be mentioned.

B. MANUSCRIPT ORGANIZATION AND FORMATTING:

Sincerely,

October 8, 2024

RE: Life Science Alliance Manuscript #LSA-2024-03052-TR

Dr. Francois El-Daher
University of Edinburgh
The Chancellor's Building
49 Little France Crescent
Edinburgh EH16 4SB
United Kingdom

Dear Dr. El-Daher,

Thank you for submitting your Research Article entitled "Microglia are essential for tissue contraction in wound closure after brain injury in zebrafish". It is a pleasure to let you know that your manuscript is now accepted for publication in Life Science Alliance. Congratulations on this interesting work.

DISTRIBUTION OF MATERIALS:

Again, congratulations on a very nice paper. I hope you found the review process to be constructive and are pleased with how the manuscript was handled editorially. We look forward to future exciting submissions from your lab.

Sincerely,
